# ALIGNING HUMAN MOTION GENERATION WITH HUMAN PERCEPTIONS

**Haoru Wang**[1*]    **Wentao Zhu**[1*]    **Luyi Miao**[1]    **Yishu Xu**[1]
**Feng Gao**[2,3]    **Qi Tian**[4]    **Yizhou Wang**[1,3,5,6]
[1]Center on Frontiers of Computing Studies, School of Compter Science, Peking University
[2]School of Arts, Peking University
[3]Inst. for Artificial Intelligence, Peking University
[4]Huawei Technologies, Ltd.
[5]Nat'l Eng. Research Center of Visual Technology, Peking University
[6]State Key Laboratory of General Artificial Intelligence, Peking University

## ABSTRACT

Human motion generation is a critical task with a wide range of applications. Achieving high realism in generated motions requires naturalness, smoothness, and plausibility. Despite rapid advancements in the field, current generation methods often fall short of these goals. Furthermore, existing evaluation metrics typically rely on ground-truth-based errors, simple heuristics, or distribution distances, which do not align well with human perceptions of motion quality. In this work, we propose a data-driven approach to bridge this gap by introducing a large-scale human perceptual evaluation dataset, `MotionPercept`, and a human motion critic model, `MotionCritic`, that capture human perceptual preferences. Our critic model offers a more accurate metric for assessing motion quality and could be readily integrated into the motion generation pipeline to enhance generation quality. Extensive experiments demonstrate the effectiveness of our approach in both evaluating and improving the quality of generated human motions by aligning with human perceptions. Code and data are publicly available at `https://motioncritic.github.io/`.

## 1 INTRODUCTION

Human motion generation is an important emerging task (Zhu et al., 2024) with wide-ranging applications, including augmented and virtual reality (AR/VR) (Yin et al., 2022; Yang et al., 2020), human-robot interaction (Nishimura et al., 2020; Gulletta et al., 2020), and digital humans (Kucherenko et al., 2019; Yi et al., 2023). Achieving high realism in generated human motions is crucial, necessitating naturalness, smoothness, and plausibility. However, current generation methods still fall short of these goals, often producing subpar results. Meanwhile, designing appropriate evaluation metrics that accurately reflect these qualities remains a significant challenge. This complexity stems from the highly non-linear and articulated nature of human motion, which must adhere to physical and bio-mechanical constraints while also avoiding visual artifacts. Effective metrics would not only facilitate the objective comparison of generated results but also have the potential to enhance generation models by addressing their shortcomings.

Existing evaluation metrics typically rely on error with pairing ground truth (GT) motion, simple heuristics, or on distribution distance with real motion manifold. The error-based metrics cannot fully reflect the performance because GT is only one reasonable possibility. The heuristics fall short in comprehensively representing motion quality. For instance, foot-ground contact metrics (Rempe et al., 2021; Tseng et al., 2023) fail to penalize twisting arm motions that violate bio-mechanical constraints. It is also infeasible to manually define all the human motion rules in a handcrafted manner. Meanwhile, distribution distance metrics like Fréchet Inception Distance (FID) (Heusel et al., 2017) do not operate on an instance level but rather assess overall distribution similarity. Consequently, they cannot identify implausible motions or provide direct supervision signals to guide the generation of higher-quality motions. Some studies (Tseng et al., 2023; Voas et al., 2023) also indicate that FID correlates poorly with user studies due to the misalignment between its distance measurement and

---

*Lead Authors.

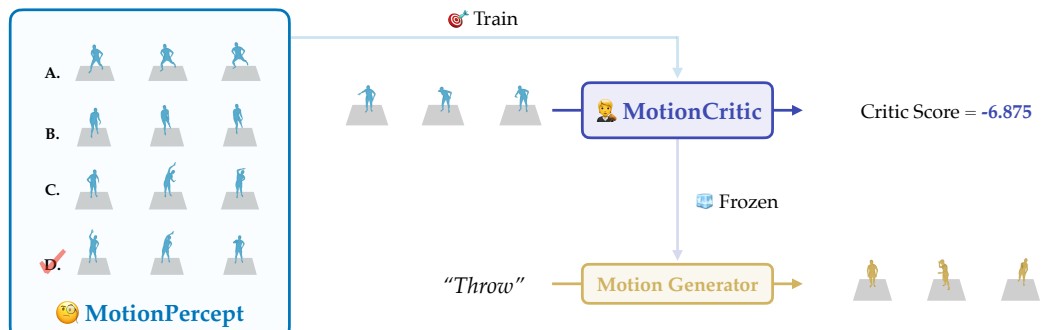

Figure 1: Framework Overview. We collect `MotionPercept`, a large-scale, human-annotated dataset for motion perceptual evaluation, where human subjects select the best quality motion in multiple-choice questions. Using this dataset, we train `MotionCritic` to automatically judge motion quality in alignment with human perceptions, offering better quality metrics. Additionally, we show that `MotionCritic` can enhance existing motion generators with minimal fine-tuning.

human perception of motion quality. Consequently, existing automatic evaluation metrics cannot effectively reflect or replace subjective user studies, hindering objective evaluation and comparison.

In light of this, we advocate the need for *automatic evaluation aligned with human perceptions*. Firstly, humans are the primary audience and interaction partners for motion generation, making their perception crucial for evaluating motion quality. Secondly, the human brain possesses specialized neural mechanisms for processing biological motion (Blakemore & Decety, 2001; Grossman et al., 2000) and is sensitive to even slightly unnatural motions (Troje, 2002; Shimada & Oki, 2012). Therefore, we explore the possibility of directly learning perceptual evaluations from humans using a data-driven approach. This method could bridge the gap between objective metrics and subjective human judgments, providing a more accurate assessment of motion quality.

First, we carefully curate a human perceptual evaluation dataset named `MotionPercept`, which contains 52563 pairs of human preference annotations on generated motions. Next, we train a human motion critic model, `MotionCritic`, that learns motion quality ratings from the collected dataset. Our critic model significantly outperforms previous metrics in terms of alignment with human perceptions. Notably, it generalizes well across different data distributions. In addition to motion evaluation, we further propose to utilize the critic model as a direct supervision signal. We demonstrate that `MotionCritic` can be seamlessly integrated into the generation training pipeline, effectively improving motion generation quality by increasing alignment with human perceptions with few steps of finetuning.

We summarize our contributions as follows: 1) We contribute `MotionPercept`, a large-scale motion perceptual evaluation dataset with manual annotations. 2) We develop `MotionCritic` which models human perceptions of motions through a data-driven approach. Extensive experiments demonstrate its superiority as an automatic human-aligned metric of motion quality. 3) We show that the proposed motion critic model could effectively serve as a supervision signal to enhance motion generation quality. Remarkably, it requires only a small number of fine-tuning steps and can be easily integrated into existing generator training pipeline in a plug-and-play manner.

## 2 RELATED WORK

### 2.1 HUMAN MOTION GENERATION

Human motion generation is a pivotal task in computer vision, computer graphics, and artificial intelligence, aiming to produce natural and realistic human pose sequences (Zhu et al., 2024). This field has seen substantial advancements with the rise of deep generative models (Kingma & Welling, 2014; Rezende & Mohamed, 2015; Goodfellow et al., 2014; Ho et al., 2020). Previous works have explored text-conditioned motion generation that transform narrative descriptions into coherent pose sequences (Tevet et al., 2023; 2022; Li et al., 2017; Petrovich et al., 2022; Lucas* et al., 2022), audio-conditioned methods that synchronize movements with rhythmic cues (Huang et al., 2021; Siyao et al.,

2022; Tseng et al., 2023), and scene-conditioned generation that integrates environmental contexts to produce contextually appropriate motions (Corona et al., 2020; Wang et al., 2021; Araújo et al., 2023). Despite significant progress, current mainstream data-driven kinematic motion generation methods sometimes produce unnatural motions that are jittery, distorted, or violate physiological and physical constraints. These issues could be attributed to the inherent uncertainty of the task, limitations of supervision signals, and dataset noises. Furthermore, evaluating generated human motions presents additional challenges. Traditional metrics like error and FID fail to capture key aspects such as fluidity and biomechanical plausibility. Some works incorporate handcrafted physical priors (Tseng et al., 2023; Rempe et al., 2021), while others propose improved distance metrics for human poses or motions (Gopalakrishnan et al., 2019; Tanke et al., 2021; Tiwari et al., 2022). However, they still face limitations in fully and accurately reflecting human evaluations of motion quality. These challenges highlight the need for metrics that better align with human perception to more effectively evaluate and improve motion generation results.

## 2.2 HUMAN PERCEPTION MODELING

Pioneer work (Zhang et al., 2018) collect human perceptual similarity dataset and propose to utilize distance in deep features as perceptual metrics. Some works (Ouyang et al., 2022; Bai et al., 2022; Yuan et al., 2023b; Hejna & Sadigh, 2024; Dong et al., 2023) in language models to explore aligning model performance with human intent by first training a reward model, then performing reinforcement learning with the reward model. Recent works (Yuan et al., 2023a; Wu et al., 2023; Lee et al., 2023) also explore utilizing human feedback to improve visual generation results. For example, ImageReward (Xu et al., 2024) propose a reward feedback learning method (ReFL) to to align text-to-image generative models with human judgements. In human motion generation, however, few studies have explored modeling human feedbacks, even though the generated motion quality is highly relevant to human perceptions. HuTuMotion (Han et al., 2024) proposes to leverage few-shot human feedback with a set of representative texts. One recent work, MoBERT (Voas et al., 2023), constructs a dataset of human ratings for generated motions. Our work differs from MoBERT in that we collect real human data on a scale tens of times larger (52.6K vs 1.4K) and use comparisons instead of ratings, which is more robust. We design the critic model to learn ratings from these comparisons automatically. Additionally, our approach could not only evaluate motion quality but also effectively improve motion generation results.

## 3 MOTIONPERCEPT: A LARGE-SCALE DATASET OF MOTION PERCEPTUAL EVALUATION

We build `MotionPercept` to capture real-human perceptual evaluations with large-scale and diverse human motion sequences. Hence, we implement a rigorous and efficient pipeline for data collection and data annotation. We also design a concensus experiment in order to examine the perceptual consistency across various human subjects.

### 3.1 MOTION DATA COLLECTION

We first collect generated human motion sequence pairs for subsequent perceptual evaluation. We utilize state-of-the-art diffusion-based motion generation method MDM (Tevet et al., 2023) and FLAME (Kim et al., 2023) to generate human motion sequences parameterized by SMPL (Loper et al., 2015). For MDM (Tevet et al., 2023), we utilize the action-to-motion model trained on HumanAct12 (Guo et al., 2020) and UESTC (Ji et al., 2018) respectively. For FLAME (Kim et al., 2023), we utilize the text-to-motion model trained on HumanML3D (Guo et al., 2022). For each group of 4 motion sequences to be annotated, we use the same condition (text prompt or action labels) while sampling different random noises, with a same length of 60 frames, 24 fps. This makes the motions similar in content while still having distinguishable differences, thereby making it easier to annotate the choices.

### 3.2 HUMAN PERCEPTUAL EVALUATION

Human perceptual evaluation is the core component of `MotionPercept`, therefore we implement a rigorous pipeline to ensure annotation quality. We first introduce the question design of the perceptual

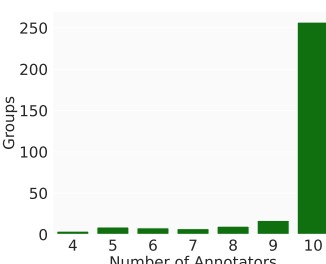 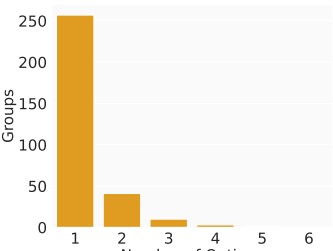 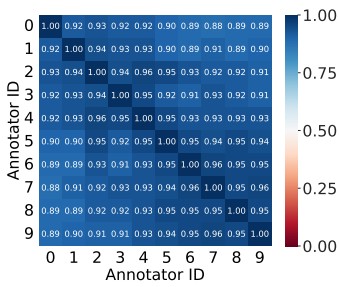

(A): Number of Annotators Select-  (B): Number of Options Selected  (C): Pairwise Agreement Ratio
ing the Most Preferred Option  Per Question  Among Annotators

Figure 2: We conduct a perceptual consensus experiment with 10 subjects on 312 multiple-choice questions, each with 6 options. (A): The distribution of the number of supporters for the most chosen option in each question. (B): Distribution of the number of options chosen by all subjects for each question. (C): Pairwise agreement ratio of all subjects.

evaluation, then describe the protocol for conducting the evaluation. Finally, we present a statistical analysis of the evaluation results.

### 3.2.1 QUESTION DESIGN

Our perceptual evaluation is designed in the form of multiple-choice questions as selection is generally easier and more robust than directly rating (Kendall, 1948; Stewart et al., 2005; Voas et al., 2023). Given a group of four motion sequence options, we instruct the annotators to select the best candidate that is most natural, visually pleasing, and free of artifacts. Specifically, we summarize the typical failure modes of the generated motions (*e.g.*, jittering, foot skating, limb distortion, penetration, *etc.*) and explicitly require the annotators to exclude these options. We provide detailed guidance with task descriptions and representative video examples to better communicate the goal to the annotators. The full guidance is presented in Appendix A.3.

While the optimal choice can be decided unambiguously in most cases, there are situations where the decision can be challenging. Therefore, we add two additional options, "all good" and "all bad", so that the annotator is not required to pick one of the motions in these cases, thereby improving overall annotation quality. Results indicate that these cases account for a small portion of the total data. We exclude these cases from our subsequent experiments. In total, we set six options for each entry: four motion candidates plus "all good" and "all bad".

### 3.2.2 PROTOCOLS

To ensure the quality of perceptual evaluation results, our annotation process consists of annotator training, annotation, and quality control. We recruit 10 annotators to perform the perceptual evaluation. Before the evaluation begins, we provide annotation guidelines to help the annotators understand the task and maintain consistent criteria. The annotators must pass a pilot test before starting the formal annotation to ensure they correctly understand the annotation requirements. Additionally, we conduct a perceptual consensus experiment to assess whether the annotation pipeline is suitable for our dataset, as discussed in Section 3.3. Finally, we implement a quality control process where the annotated data is reviewed by an expert quality inspector. During the annotation process, we continuously monitor the quality of each batch of data. For each batch, we randomly sample 10% of the data for quality inspection. The consistency between the sampled data and the expert's annotations must exceed 90%; otherwise, the entire batch will be re-annotated. Complete protocols are detailed in Appendix A.

### 3.3 ANALYSIS

In total, we collect annotations for 18260 multiple-choice questions covering 73040 unique motions, significantly surpassing previous work (Voas et al., 2023) (1400 motions). We further investigate the following two questions:

1. Based on our experimental setup, can the subjects confidently select the suitable options from the choices provided?

2. Is there a significant difference in perceptual preferences among different subjects, or are they well-aligned?

For the first question, we calculate the proportion of cases where a choice could not be made (including "all good" and "all bad"), and find a total of 418 such groups (2.29%). The result indicates that most of the time subjects can make a definite judgment, demonstrating the validity of our protocol design.

For the second question, we conduct a perceptual consensus experiment where all 10 subjects perform perceptual evaluation independently on 312 groups of randomly selected data. We calculate their pairwise and overall consistency in choices. Figures 2(A) and 2(B) show that for most questions (82.37%), all 10 subjects make the unanimous decision. Figure 2(C) reveals that all 10 subjects exhibit high pairwise agreement (90%). These results indicate a high level of consistency in perceptual judgments of human motion among different human subjects. This not only validates the rationality of our perceptual evaluation pipeline but also inspires us to train machine learning models to emulate this consistent judgment capability.

## 4 MOTIONCRITIC: ADVANCING MOTION GENERATION WITH PERCEPTUAL ALIGNMENT

Based on `MotionPercept`, we develop a human motion critic model, `MotionCritic`, to emulate the perceptual judgment capabilities of human subjects regarding human motion. We first present the problem formulation and training approach of the critic model, and then explain how to use the critic model for optimizing motion generation.

### 4.1 PROBLEM FORMULATION

We formulate the problem as follows: given an input human motion sequence $\mathbf{x}$, we assume there is an implicit human perception model $\mathcal{H}$ that rates the motion quality $\mathcal{H}(\mathbf{x})$, where a higher rate indicates better quality. We aim to build a computational critic model $\mathcal{C}$ that best aligns with $\mathcal{H}$. Since $\mathcal{H}$ is not explicitly available, we take a data-driven approach. We obtain the human perceptual evaluation dataset $\mathcal{D}$ containing multiple pairs of samples $(\mathbf{x}^{(i)}, \mathbf{x}^{(j)})$. Our training objective is to train the model $\mathcal{C}$ using the dataset $\mathcal{D}$ so that it approximates the human perception model $\mathcal{H}$ as closely as possible. Specifically, we want the model prediction $\mathcal{C}(\mathbf{x}^{(i)}) > \mathcal{C}(\mathbf{x}^{(j)})$ if and only if $\mathcal{H}(\mathbf{x}^{(i)}) > \mathcal{H}(\mathbf{x}^{(j)})$. Based on the Bradley-Terry model (Bradley & Terry, 1952; Hunter, 2004), the overall training objective could be written as maximizing the joint probabilities that the model $\mathcal{C}$ makes judgments consistent with $\mathcal{H}$ for each pair of samples in the dataset $\mathcal{D}$:

$$\arg\max_{\mathcal{C}} \mathbb{E}_{(\mathbf{x}^{(i)}, \mathbf{x}^{(j)}) \sim \mathcal{D}} \left[ \log \sigma \left( (\mathcal{C}(\mathbf{x}^{(i)}) - \mathcal{C}(\mathbf{x}^{(j)})) \cdot (\mathcal{H}(\mathbf{x}^{(i)}) - \mathcal{H}(\mathbf{x}^{(j)})) \right) \right], \tag{1}$$

where $\sigma$ is the sigmoid function.

### 4.2 HUMAN MOTION CRITIC MODEL

In practice, we represent human motion by $\mathbf{x} \in \mathbb{R}^{L \times J \times D}$ where $L$ denotes the sequence length, $J$ denotes the number of body joints, and $D$ denotes parameter dimensions. We implement the critic model $\mathcal{C}$ as a neural network that maps the high-dimensional motion parameters to a scalar $s$. We draw pairwise comparison annotations from the collected dataset, where $\mathbf{x}^{(h)}$ is the better instance and $\mathbf{x}^{(l)}$ is the worse. The perceptual alignment loss is thus given by:

$$\mathcal{L}_{\text{Percept}} = -\mathbb{E}_{(\mathbf{x}^{(h)}, \mathbf{x}^{(l)}) \sim \mathcal{D}} \left[ \log \sigma \left( \mathcal{C}(\mathbf{x}^{(h)}) - \mathcal{C}(\mathbf{x}^{(l)}) \right) \right]. \tag{2}$$

### 4.3 MOTION GENERATION WITH CRITIC MODEL SUPERVISION

Additionally, we explore to utilize the learned human perceptual prior of $\mathcal{C}$ not only for evaluating generated motions, but also improving them. We demonstrate that our motion critic model could

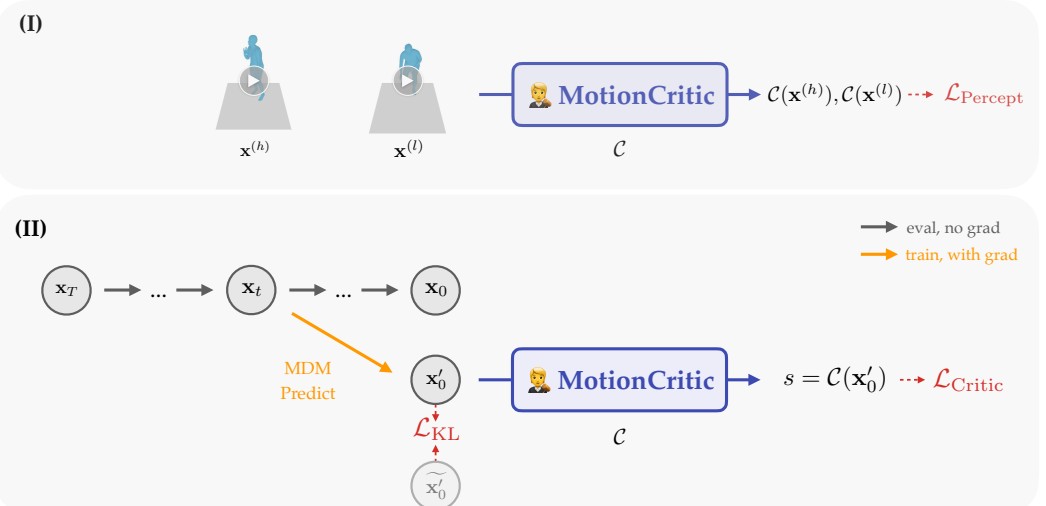

Figure 3: (I) Critic model training process. We sample human motion pairs $\mathbf{x}^{(h)}, \mathbf{x}^{(l)}$ annotated with human preferences, upon which the critic model produces score pairs. We use perceptual alignment loss $L_{\text{Percept}}$ to learn from the human perceptions. (II) Motion generation with critic model supervision. We intercept MDM sampling process at random timestep $t$ and perform single-step prediction. Critic model computes the score $s$ based on the generated motion $\mathbf{x}_0'$, which is further used to calculate motion critic loss $L_{\text{Critic}}$. KL loss $L_{\text{KL}}$ is introduced between $\mathbf{x}_0'$ and last-time generation result $\widetilde{\mathbf{x}_0}'$.

be integrated into state-of-the-art diffusion-based motion generation approaches with ease by using MDM (Tevet et al., 2023) as an example. The forward diffusion is modeled as a Markov noising process $\{\mathbf{x}_t\}_{t=0}^T$ where $\mathbf{x}_0$ is drawn from the data distribution, and

$$q(\mathbf{x}_t|\mathbf{x}_{t-1}) = \mathcal{N}(\sqrt{\alpha_t}\mathbf{x}_{t-1}, (1 - \alpha_t)I), \tag{3}$$

where $\alpha_t \in (0, 1)$ are constant hyper-parameters. When $\alpha_t$ is small enough, it's reasonable to approximate $\mathbf{x}_T \sim \mathcal{N}(0, I)$, allow sampling $\mathbf{x}_T$ from random noise to begin our denoising process.

Given an MDM model $\mathcal{M}$ with pre-trained parameters $\theta_0$, we fine-tune to improve its alignment with a pre-trained critic model $\mathcal{C}$. We develop a lightweight perceptual-aligned fine-tuning approach based on ReFL (Xu et al., 2024). Notably, in order to utilize the critic model in a plug-and-play manner, we keep the MDM training step and objective $\mathcal{L}_{\text{MDM}}$ unchanged. Of each fine-tuning iteration, the original MDM loss is computed using text labels and ground-truth motions as the initial step. This loss is first utilized to update the pre-trained model weights accordingly. After that, we simply add another optimization step with critic model supervision. Structure of this optimization step is shown in Figure 3, and detailed as follows:

---

**Algorithm 1** Fine-tuning Motion Generation with `MotionCritic`

1: **Dataset:** Action-label set $\widetilde{\mathcal{D}} = \{(\text{label}_i, \text{mot}_i)\}$
2: **Input:** MDM model $\mathcal{M}_{\theta_0}$, Critic model $\mathcal{C}$, Critic-to-loss map function $\phi$, Critic loss scale $\lambda$, KL loss scale $\mu$
3: **for** $(\text{label}_i, \text{mot}_i) \in \widetilde{\mathcal{D}}$ **do**
4:     $\theta_i \leftarrow \theta_i$            $\triangleright$ Update MDM$_{\theta_i}$ with $\mathcal{L}_{\text{MDM}}$
5:     $t \leftarrow \text{rand}(T_1, T_2)$    $\triangleright$ Pick a random time step $t \in [T_1, T_2]$
6:     $\mathbf{x}_T \sim \mathcal{N}(0, I)$                  $\triangleright$ Sample noise
7:     **for** $j = T, ..., t + 1$ **do**
8:         **no grad:** $\mathbf{x}_{j-1} \leftarrow \mathcal{M}_{\theta_i}\{\mathbf{x}_j\}$
9:     **end for**
10:    **with grad:** $\widetilde{\mathbf{x}_0}' \leftarrow \mathcal{M}_{\theta_i}\{\mathbf{x}_t\}$
11:    **if** $\widetilde{\mathbf{x}_0}'$ is not None **then**
12:       $\mathcal{L}_{\text{KL}} \leftarrow \mu\text{KL}(\widetilde{\mathbf{x}_0}', \mathbf{x}_0')$    $\triangleright$ KL loss with previous $\mathbf{x}_0'$
13:    **end if**
14:    $\mathcal{L}_{\text{Critic}} \leftarrow \lambda\phi(\mathcal{C}(\mathbf{x}_0'))$         $\triangleright$ Critic loss
15:    $\theta_{i+1} \leftarrow \theta_i$   $\triangleright$ Update $\mathcal{M}_{\theta_i}$ with $\mathcal{L}_{\text{Critic}}$ and $\mathcal{L}_{\text{KL}}$
16:    $\widetilde{\mathbf{x}_0}' \leftarrow \mathbf{x}_0'$        $\triangleright$ Save $\mathbf{x}_0'$ for next-step $\mathcal{L}_{\text{KL}}$
17: **end for**

---

We begin with sampling a Gaussian noise $\mathbf{x}_T$ and performing gradient-free denoising steps until $\mathbf{x}_t$, where $t \in [T_1, T_2]$ is randomly selected in denoising steps. Gradient interception range $[T_1, T_2]$

are hyperparameters, with selection principles outlined in Appendix C.1. After reaching step $t$, a single-step denoising with gradient is performed directly to predict $\mathbf{x}_0'$ from $\mathbf{x}_t$. Upon the predicted motion $\mathbf{x}_0'$, we compute its critic score $s = \mathcal{C}(\mathbf{x}_0')$, and use this score output to compute the motion critic loss. We formulate our critic loss as follows:

$$\mathcal{L}_{\text{Critic}} = \mathbb{E}_{y_i \sim \mathcal{Y}} \left[ \phi(\mathcal{C}(\mathbf{x}_0')) \right], \tag{4}$$

where $\phi(s) = -\sigma(\tau - s))$ is a critic-to-loss mapping function, $\tau$ being a constant threshold for shifting the critic value, and $\sigma$ being sigmoid function.

We further introduce a Kullback-Leibler (KL) divergence regularization to prevent $\mathcal{M}$ from moving substantially away from the conditional motion generation task. We formulate our KL loss as follows:

$$\mathcal{L}_{\text{KL}} = \mathbb{E}_{y_i \sim \mathcal{Y}} \left[ D_{\text{KL}} \left( p(\mathbf{x}_0') \| p(\widetilde{\mathbf{x}_0'}) \right) \right]. \tag{5}$$

where $\widetilde{\mathbf{x}_0'}$ being $\mathbf{x}_0'$ of the previous iteration. Overall, our fine-tuning loss is given by

$$\mathcal{L}_{\text{FT}} = \mathcal{L}_{\text{MDM}} + \lambda \mathcal{L}_{\text{Critic}} + \mu \mathcal{L}_{\text{KL}}. \tag{6}$$

where $\lambda$ and $\mu$ are re-scaling weights for loss balancing. Detailed algorithm workflow is shown in Algorithm 1.

## 5 EXPERIMENT

### 5.1 IMPLEMENTATION DETAILS

**Critic Model.**    We train our critic model using the MDM subset in `MotionPercept`. We convert each multiple-choice question into three ordered preference pairs, which results in 46740 pairs for training and 5823 pairs for testing. We parameterize motion sequences with SMPL (Loper et al., 2015), including 24 axis-angle rotations, and global root translation. We implement the critic model with DSTformer (Zhu et al., 2023) backbone with 3 layers and 8 attention heads. We apply temporal average pooling on encoded motion embeddings followed by an MLP with a hidden layer of 1024 channels to predict a single scalar score. We train the critic model for 150 epochs with a batch size of 64 and a learning rate starting at 2e-3, decreasing with a 0.995 exponential learning rate decay.

**Fine-tuning.**    We use MDM (Tevet et al., 2023) model trained on HumanAct12 (Guo et al., 2020) as our baseline, which utilizes 1000 DDPM denoising steps. We load the checkpoint trained for 350000 iterations and fine-tune for 800 iterations, with a batch size of 64 and learning rate 1e-5. We fine-tune with critic clipping threshold $\tau = 12.0$, critic re-weight scale $\lambda =$1e-3, and KL loss re-weight scale $\mu = 1.0$. We set the step sampling range $[T_1, T_2] = [700, 900]$. Details of the setup are shown at Appendix C.1.

### 5.2 MOTIONCRITIC AS MOTION QUALITY METRIC

We first evaluate whether the proposed critic model could serve as an effective motion quality metric. Specifically, we are interested in the following research questions:

1. How does `MotionCritic` align with human perceptual evaluations?
2. Could `MotionCritic` generalize to different data distributions?

To investigate the first question, we evaluate the performance of our critic model on a held-out test set and compare it with existing motion quality metrics as follows:

- **Distance-based metrics**, including Root Average Error (Root AVE), Root Absolute Error (Root AE), Joint Average Error (Joint AVE), and Joint Absolute Error (Joint AE). These metrics involve directly computing the distance between the generated motion and a pairing GT with the same condition. Furthermore, methods like PoseNDF (Tiwari et al., 2022), Normalized Power Spectrum Similarity (NPSS)(Gopalakrishnan et al., 2019), and Normalized Directional Motion Similarity (NDMS)(Tanke et al., 2021) offer more advanced approaches to measuring distances between motions.

Table 1: Quantitative comparison of motion evaluation metrics on MDM and FLAME testsets of `MotionPercept`.

| Metric | MDM | | FLAME | |
|---|---|---|---|---|
| | Acc. (%) ↑ | Log Loss ↓ | Acc. (%) ↑ | Log Loss ↓ |
| Root AVE | 59.47 | 0.6891 | 48.42 | 0.6984 |
| Root AE | 61.79 | 0.6798 | 59.54 | 0.6711 |
| Joint AVE | 56.77 | 0.6889 | 44.61 | 0.6973 |
| Joint AE | 62.73 | 0.6794 | 58.37 | 0.6891 |
| Jerk | 65.48 | 0.7516 | 65.84 | 0.5984 |
| Acceleration (Yang et al., 2023) | 64.26 | 0.7792 | 66.67 | 0.6919 |
| Person-Ground Contact (Rempe et al., 2021) | 71.78 | 0.7260 | 69.82 | 0.7243 |
| Foot-Floor Penetration (Rempe et al., 2021) | 53.61 | 0.6939 | 55.56 | 0.6906 |
| Physical Foot Contact (Tseng et al., 2023) | 64.79 | 0.6926 | 66.00 | 0.6930 |
| PoseNDF (Tiwari et al., 2022) | 55.13 | 0.6930 | 53.07 | 0.6931 |
| NPSS (Gopalakrishnan et al., 2019) | 52.37 | 0.6911 | 52.07 | 0.6944 |
| NDMS (Tanke et al., 2021) | 63.92 | 0.6519 | 63.35 | 0.6301 |
| MoBERT (Voas et al., 2023) | 49.40 | 0.6931 | 52.40 | 0.6932 |
| `MotionCritic` (Ours) | **85.07** | **0.5486** | **81.43** | **0.5758** |

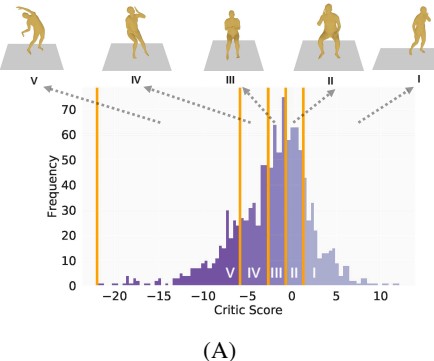
(A)

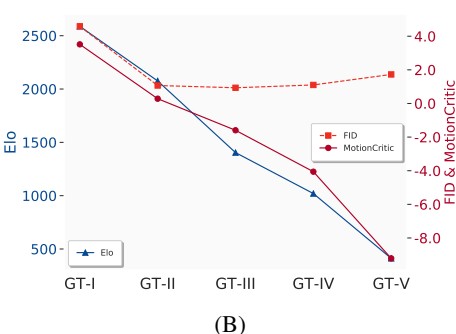
(B)

Figure 4: We group HumanAct12(Guo et al., 2020) GT test set into 5 subsets, and compare their qualities. (A): GT-I to GT-V subsets split based on critic scores from high to low. (B): Elo ratings from user study, FID and average critic scores of different GT subsets.

- **Heuristic metrics**, including acceleration (Rempe et al., 2021; Yang et al., 2023), Person-Ground Contact (Rempe et al., 2021), Foot-Floor Penetration (Rempe et al., 2021), and Physical Foot Contact (PFC) (Tseng et al., 2023). These metrics does not compare against GT; instead, they implement intuitive rule-based evluations. For example, PFC models the relationship between center of mass acceleration and foot-ground contact.

- **Learning-based metrics**. Prior work MoBERT (Voas et al., 2023) proposes to evaluate motion quality with a motion feature extractor and SVR Regression.

Note that distribution-based metrics (*e.g.* FID) could not compare quality of individual motion sequences, and the comparison can be found in subsequent experiments. For each metric, we calculate the percentage they align with GT annotations (accuracy) and also their probabilistic distribution distance with GT annotations (log loss). We use the softmax function to convert the scores to probabilities (taking the opposite before softmax for metrics where smaller is better). Table 1 demonstrates that our critic model significantly outperforms previous metrics. These results not only validate the effectiveness of learning from large-scale human perceptual evaluations but also prove that our critic model can serve as a more comprehensive and robust metric for assessing motion quality.

Furthermore, to investigate the second question, we test the critic model on data outside of the training distributions. We collect a standalone test set from 804 motions with a different motion generation algorithm, FLAME (Li et al., 2017), and perform perceptual evaluation with a different human subject. Note that this model is trained on a different dataset (Guo et al., 2022) with the model

Figure 5: Model performance during fine-tuning process. (A): User study win rates (row vs column) with different fine-tuned model steps. (B): Elo ratings from user study, FID and average critic scores in the fine-tuning process.

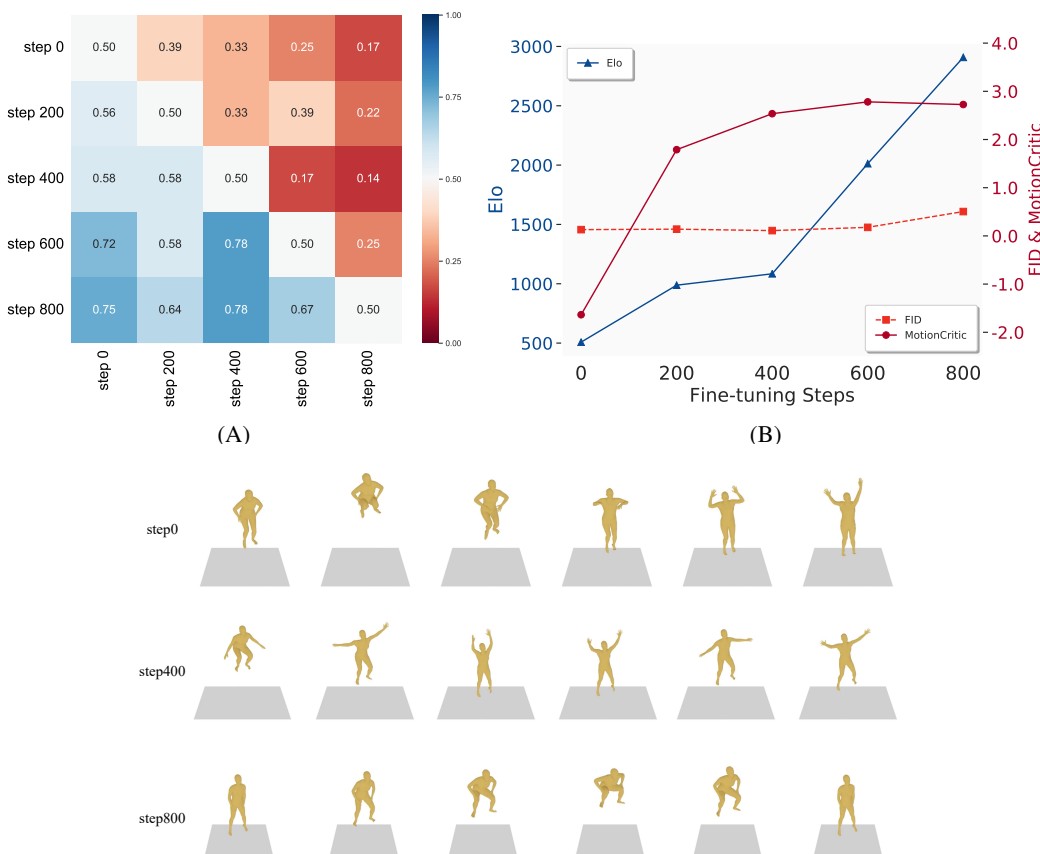

Figure 6: Motion generation results from different fine-tuning steps.

used to generate critic model training data, which means the action categories have large variations. The results in Table 1 further shows that our critic model could well generalize to the new test set, indicating its efficacy in evaluating different generation algorithms and unseen motion contents.

Additionally, we test the generalization of our critic model on the real GT motion distribution. Figure 4(A) illustrates the critic score distribution of HumanAct12 (Guo et al., 2020) test set. We group the 1190 GT motions into 5 groups based on their critic scores, evenly distributed from highest to lowest. We compare the average critic score between the groups with distribution-based metric FID and user study. The user study is conducted by another 5 users different from previous annotators, comparing motion pairs sampled from each groups and then computing Elo rating (Elo, 1978; Tseng et al., 2023) for each group. Figure 4(B) clearly indicates that the critic score aligns well with human preferences, while FID does not. Notably, we discover that the outliers with small critic values (group V) are indeed artifacts within the dataset. Please refer to to Appendix D for details in user study, and the supplementary materials for video results.

The results indicate that our critic model can also generalize to the GT motion manifold, even though the model has never been trained on it. It also highlights the potential of using our critic model as a tool for dataset diagnosis (*e.g.*, discover failure modes). Please also refer to Appendix B for additional results, where we provide more discussions on `MotionCritic`, including failure modes, out-of-distribution tests, the impact of different motion lengths, and its relationship with other metrics.

## 5.3 MotionCritic as Training Supervision

Furthermore, we investigate whether our critic model can also serve as an effective supervision signal. Specifically, we fine-tune a pre-trained motion generator (Tevet et al., 2023) with the proposed

Table 2: Comparison of motion generation metrics at different fine-tuning steps.

| Step | FID↓ | PFC↓ | MotionCritic↑ | Accuracy↑ | Diversity↑ | Multimodality↑ |
|------|------|------|------|------|------|------|
| 0 | 0.13 | 0.00095 | -1.64 | 0.98 | 6.66 | 2.15 |
| 100 | 0.17 | **0.00086** | 1.83 | 0.98 | 6.62 | 2.32 |
| 200 | 0.14 | 0.0010 | 1.79 | 0.98 | 6.58 | 2.46 |
| 300 | 0.14 | 0.0010 | 2.35 | 0.98 | 6.56 | 2.36 |
| 400 | **0.11** | 0.0012 | 2.54 | 0.98 | 6.61 | 2.63 |
| 500 | 0.17 | 0.0013 | 2.61 | 0.98 | 6.62 | 2.36 |
| 600 | 0.18 | 0.0013 | 2.78 | **0.98** | **6.68** | 2.42 |
| 700 | 0.54 | 0.0017 | **3.45** | 0.92 | 6.53 | 2.43 |
| 800 | 0.50 | 0.0012 | 2.73 | 0.88 | 6.57 | **3.04** |

framework, and evaluate on HumanAct12 (Guo et al., 2020) test set every 200 steps. Additionally, we conduct a standalone user study by comparing motion pairs generated at different fine-tuning steps and compute the Elo Rating (Elo, 1978; Tseng et al., 2023). Figure 5 reveals that as fine-tuning progresses, the motion quality consistently improves according to the user study, in line with the training objective of increasing the critic score. The user study is conducted with new evaluators which are not participated in dataset annotation. Plese refer to Appendix D for details of the user study. We also present a visualization comparison in Figure 6. We discover that as fine-tuning progresses, unreasonable human motions such as jittering, twisting, and floating significantly decrease. Please refer to the supplementary materials for video comparisons.

In addition, we examine the impact of fine-tuning with `MotionCritic` on other motion evaluation metrics. We track changes in various metrics, including *Accuracy*, *Diversity*, and *Multimodality* (Tevet et al., 2023), over the course of fine-tuning from 0 to 800 steps. These metrics capture different aspects of motion generation, such as text alignment and richness. Table 2 shows that our critic model aligns more closely with human judgment compared to motion quality metrics like FID and PFC. Moreover, fine-tuning with `MotionCritic` does not conflict with other key metrics. For example, at 600 steps, we observe improvements in *Accuracy*, *Diversity*, and *Multimodality*, along with a significant increase in user preference compared to the baseline (step = 0). In conclusion, `MotionCritic` provides a more holistic evaluation of the intrinsic quality of the motion compared to other metrics in this domain. At the same time, it complements other metrics that focus on different aspects, such as text-motion consistency and diversity, resulting in a more comprehensive and objective evaluation.

The results also demonstrate that our fine-tuning process requires only hundreds of iterations to take effect, significantly improving the perceptual quality of the model. Compared to the 350K pre-training steps, this accounts for only 0.23% of the training cost. This further demonstrates the advantages of our proposed framework in using a perceptually-aligned critic model to fine-tune the motion generation model, not only improving quality but also being lightweight and efficient. Please refer to Appendix C for more details and analysis of fine-tuning.

## 6 CONCLUSION

In conclusion, our work bridges the important gap in human motion generation between objective metrics and human perceptual evaluations by introducing a data-driven framework with `MotionPercept` and `MotionCritic`. This paradigm not only offers a more comprehensive metrics of motion quality but could also improve the generation results by aligning with human preferences. We hope this work could contribute to more objective evaluations of motion generation methods and results.

To prevent potential misuse or misunderstanding, it is important to note that the critic model should be used either as an standalone evaluation metric or as a loss function for fine-tuning, but not for both simultaneously. It's also worth noticing that the energy landscape of `MotionCritic` is not smooth, and `MotionCritic` should not be interpreted as a distance measure. Discussions are detailed in Appendix B.3. Another limitation of our approach is its primary focus on perceptual metrics without explicitly simulating physical and biomechanical plausibility, which could be explored in future work. Future research could also investigate more fine-grained perceptual evaluation methods to obtain rich human feedback on motion quality like (Liang et al., 2024).

## 7 ACKNOWLEDGMENTS

This work was supported by the National Key Research and Development Program of China (No. 2022YFF0902302) and NSFC-6247070125. We extend our gratitude to the reviewers for their insightful comments and valuable discussions.

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

# Part II

# Appendix

## A    DETAILS ON MOTIONPERCEPT

### A.1    DATA GENERATION

We use synthetic data for annotating MotionPercept, because we hypothesis that using options from the same synthetic distribution would be most effective for developing a robust and generalizable critic model. Incorporating GT data, which may be of significantly higher quality, could make it too easy for the model to differentiate between options, potentially reducing its discrimination power. This practice aligns with approaches in training reward models for language models like GPT (Ouyang et al., 2022), where reward models are trained using human feedback on model-generated outputs. We note that the critic model trained subsequently produced reasonable results on out-of-distribution datasets including GT motion, demonstrating its ability to generalize beyond the synthetic data.

We utilize the prompts from HumanAct12 (Guo et al., 2020), UESTC (Ji et al., 2018) and HumanML3D (Guo et al., 2022) for generating the motion candidates. Specifically, we use the 12 action labels from HumanAct12 (Guo et al., 2020) (shown in Table 3) and the 40 categories of aerobic exercise description from UESTC (Ji et al., 2018) (shown in Table 4) for the MDM (Tevet et al., 2023) model. We randomly select texts from HumanML3D (Guo et al., 2022) test set as prompts for the FLAME (Li et al., 2017) model.

| HumanAct12 (Guo et al., 2020) Action Labels | |
|---|---|
| warm up | walk |
| run | jump |
| drink | lift dumbbell |
| sit | eat |
| turn steering wheel | phone |
| boxing | throw |

Table 3: 12 action labels from HumanAct12 (Guo et al., 2020).

| UESTC (Ji et al., 2018) Action Labels | |
|---|---|
| punching and knee lifting | marking time and knee lifting |
| jumping-jack | squatting |
| forward-lunging | left-lunging |
| left-stretching | raising-hand-and-jumping |
| left-kicking | rotation-clapping |
| front-raising | pulling-chest-expanders |
| punching | wrist-circling |
| single-dumbbell-raising | shoulder-raising |
| elbow-circling | dumbbell-one-arm-shoulder-pressing |
| arm-circling | dumbbell-shrugging |
| pinching-back | head-anticlockwise-circling |
| shoulder-abduction | deltoid-muscle-stretching |
| straight-forward-flexion | spinal-stretching |
| dumbbell-side-bend | standing-opposite-elbow-to-knee-crunch |
| standing-rotation | overhead-stretching |
| upper-back-stretching | knee-to-chest |
| knee-circling | alternate-knee-lifting |
| bent-over-twist | rope-skipping |
| standing-toe-touches | standing-gastrocnemius-calf |
| single-leg-lateral-hopping | high-knees-running |

Table 4: 40 action labels from UESTC (Ji et al., 2018).

## A.2 ANNOTATION MANAGEMENT

We recruit 10 annotators for this task, and data entries are randomly allocated to them. We provide detailed guidelines to annotators. We evaluate the annotation result by spot check. We randomly select 10% of all data to inspect the annotation results according to guidelines and calculate the proportion of unqualified data entries. If the unqualified proportion is less than 10%, the results are considered to be acceptable. All the unqualified data entries will be re-annotated. We will update the guidelines during annotation based on spot check feedback, and annotators will study the new guidelines.

## A.3 ANNOTATION DESIGN

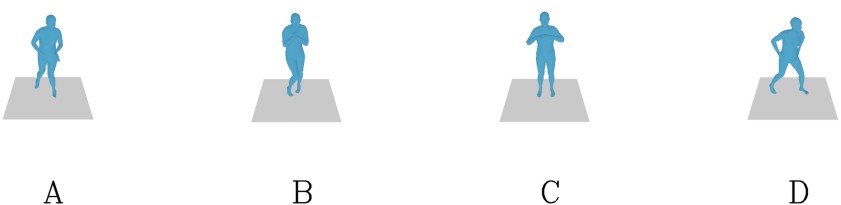

Figure 7: An example of raw data entry before annotation.

We generate four motions from the same prompt for each data entry, as shown in Fig 7. The prompts are hidden during the annotation process. Annotators are required to select either the best or the worst motion for data entries generated by MDM (Tevet et al., 2023) and FLAME (Kim et al., 2023). MDM (Tevet et al., 2023) exhibits better motion diversity but lacks stability, so annotators are instructed to select the best motion. Conversely, FLAME (Kim et al., 2023) demonstrates better stability but lacks diversity, so annotators are instructed to select the worst motion for these entries.

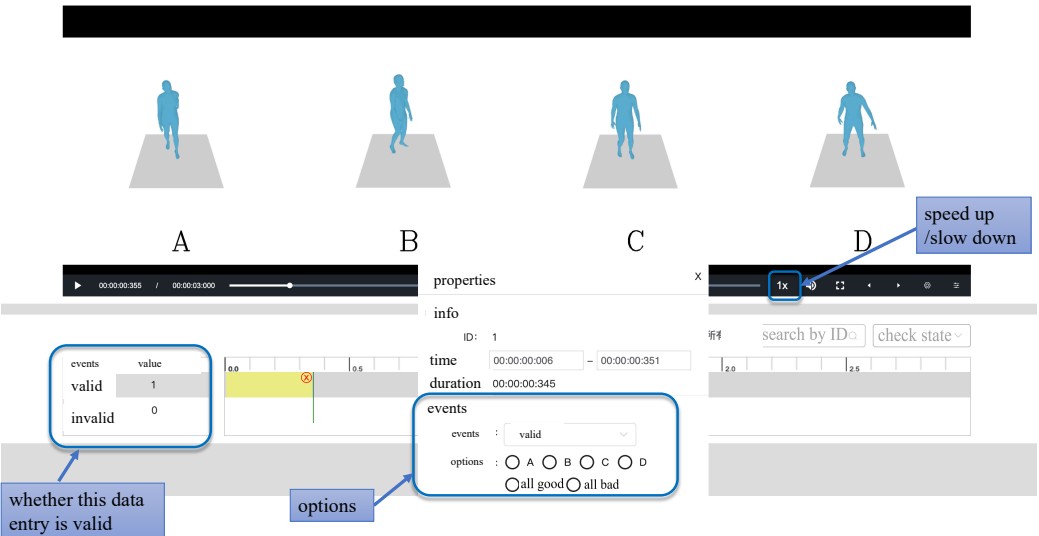

Figure 8: Our annotation platform.

## A.4 ANNOTATION GUIDANCE DOCUMENTATION

We provide a detailed annotation document to explain the annotation process. The annotation platform is shown in Fig 8.

**Introduction**    Each data entry to be annotated consists of four videos, as shown in Fig 7. Each video is approximately three seconds long, with all four videos playing simultaneously and concatenated into one video.

**Requirements**    Each set of videos has six options: A, B, C, D, "all are good," and "all are bad." Annotators should select the most natural and reasonable video for each data entry. If one option stands out as the best, select that option. If all actions seem equally good or equally bad, choose "all are good" or "all are bad." Text prompts will be hidden during annotation.

**Video Examples**    We provide annotators with examples if what kinds of motions are unnatural and unaccepetable:

1. Body pose is unnatural, including hands, feet and so on.
2. Human motion violates physiological constraints.
3. Human motion is erratic or severely stutters.
4. Human body collides, such as hands fully embedded into leg.
5. Human body is severely tilted, to the point of losing balance.
6. Human body appears to be drifting instead of walking.

Examples of these problems are shown in Fig 9.

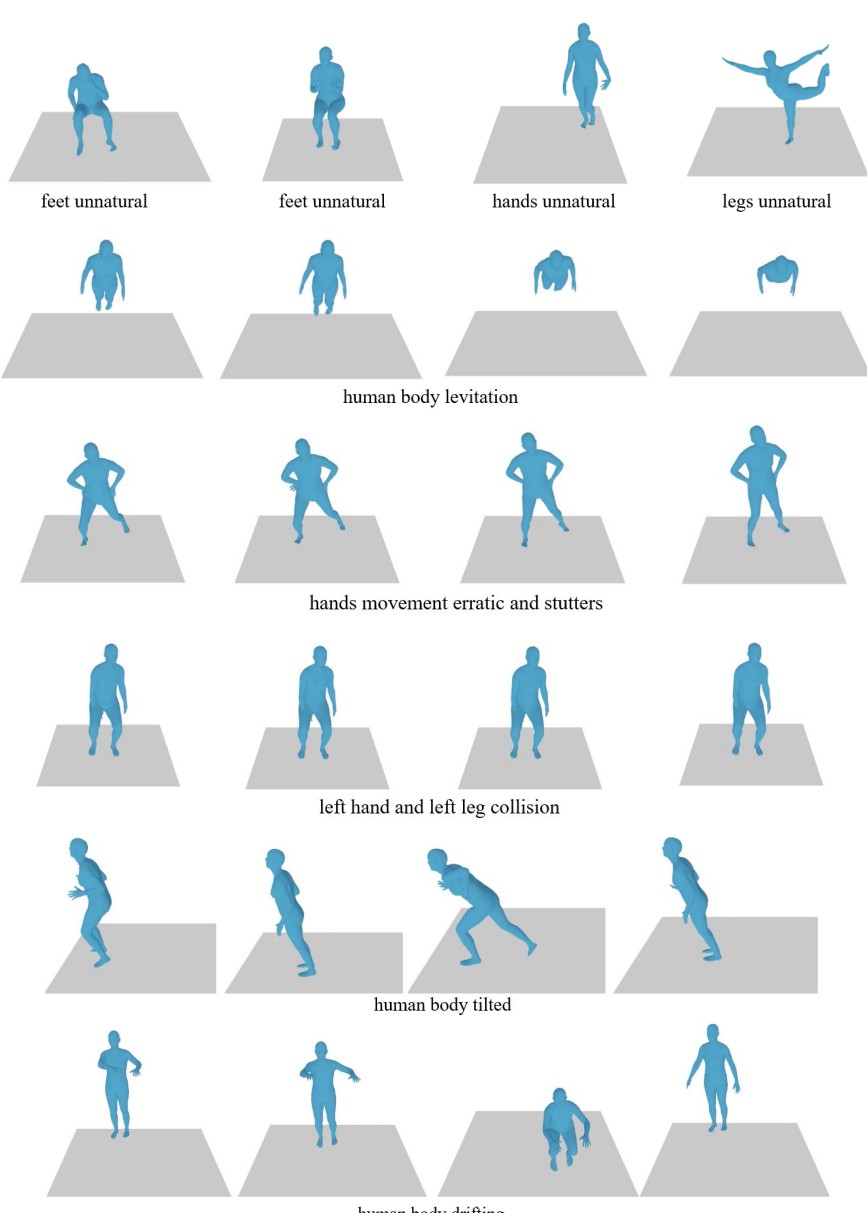

Figure 9: Representative examples of options to be excluded.

## B DETAILS ON MOTIONCRITIC: AS MOTION QUALITY METRIC

### B.1 IMPLEMENTATION DETAILS

**Data Pre-processing.** Each multiple-choice question is divided into three ordered preference pairs. Motion sequences are parameterized using SMPL (Loper et al., 2015), which includes 24 axis-angle rotations and one global root translation.

**Training and Evaluation.** We train the critic model from scratch using the DSTformer (Zhu et al., 2023) backbone with 3 layers and 8 attention heads on MotionPercept. To ensure robustness, we train our model for multiple times and report the error bars, considering variations such as the random seed across multiple runs. Evaluation results, detailing action-label splits, are presented in the following two tables. Our MotionCritic gets the best results and can robustly score different types of human motions.

| Metric | Warm. | Walk | Run | Jump | Drink | Lift. | Sit | Eat | Turn. | Phone | Box. | Throw | Avg. |
|---|---|---|---|---|---|---|---|---|---|---|---|---|---|
| Root AVE | 57.6 | 47.3 | 56.8 | 62.7 | 59.5 | 46.3 | 37.9 | 64.5 | 54.1 | 0.62 | 51.6 | 53.7 | 59.5 |
| Root AE | 70.1 | 70.0 | 69.7 | 57.2 | 70.0 | 49.8 | 52.5 | 61.2 | 63.2 | 61.7 | 52.7 | 55.2 | 61.8 |
| Joint AVE | 42.0 | 52.2 | 50.4 | 64.7 | 53.2 | 50.2 | 42.4 | 48.6 | 51.9 | 55.7 | 48.4 | 45.2 | 56.8 |
| Joint AE | 63.6 | 69.1 | 75.2 | 55.7 | 59.9 | 41.1 | 51.4 | 66.7 | 59.3 | 60.6 | 53.5 | 54.1 | 62.7 |
| Acceleration (Yang et al., 2023) | 66.7 | 78.0 | 61.6 | 53.0 | 65.4 | 62.4 | 82.6 | 61.6 | 51.1 | 59.3 | 69.5 | 61.7 | 64.3 |
| Person-Ground Contact (Rempe et al., 2021) | 69.8 | 70.1 | 70.2 | 66.0 | 72.8 | 71.8 | 90.1 | 76.9 | 70.3 | 67.9 | 62.6 | 63.2 | 71.8 |
| Foot-Floor Penetration (Rempe et al., 2021) | 47.1 | 52.4 | 52.7 | 55.7 | 48.5 | 56.8 | 59.9 | 52.4 | 50.4 | 55.7 | 52.8 | 53.3 | 53.6 |
| Physical Foot Contact (Tseng et al., 2023) | 80.5 | 77.8 | 73.1 | 51.7 | 67.5 | 57.9 | 78.5 | 63.4 | 53.2 | 65.5 | 68.6 | 68.5 | 64.8 |
| MoBERT (Voas et al., 2023) | 67.4 | 68.4 | 44.8 | 37.4 | 70.0 | 18.9 | 43.6 | 49.9 | 65.2 | 25.6 | 56.1 | 44.8 | 49.4 |
| MotionCritic (Ours) | $90.6_{\pm0.2}$ | $94.2_{\pm0.4}$ | $91.9_{\pm0.3}$ | $90.6_{\pm0.1}$ | $83.9_{\pm1.3}$ | $85.3_{\pm0.7}$ | $86.0_{\pm0.7}$ | $78.3_{\pm1.4}$ | $79.8_{\pm0.9}$ | $85.1_{\pm0.6}$ | $86.6_{\pm0.4}$ | $82.5_{\pm0.3}$ | $85.1_{\pm0.5}$ |

Table 5: Accuracy comparison of motion evaluation metrics on HumanAct12 action classes(%).

| Metric | Warm. | Walk | Run | Jump | Drink | Lift. | Sit | Eat | Turn. | Phone | Box. | Throw | Avg. |
|---|---|---|---|---|---|---|---|---|---|---|---|---|---|
| Root AVE | 0.69 | 0.69 | 0.69 | 0.69 | 0.69 | 0.69 | 0.69 | 0.69 | 0.69 | 0.69 | 0.69 | 0.69 | 0.69 |
| Root AE | 0.68 | 0.66 | 0.67 | 0.68 | 0.68 | 0.69 | 0.70 | 0.69 | 0.69 | 0.69 | 0.69 | 0.69 | 0.68 |
| Joint AVE | 0.69 | 0.69 | 0.69 | 0.69 | 0.69 | 0.69 | 0.69 | 0.69 | 0.69 | 0.69 | 0.69 | 0.69 | 0.69 |
| Joint AE | 0.68 | 0.67 | 0.67 | 0.69 | 0.68 | 0.70 | 0.70 | 0.69 | 0.69 | 0.68 | 0.68 | 0.69 | 0.68 |
| Acceleration (Yang et al., 2023) | 0.70 | 0.59 | 0.88 | 1.5 | 0.60 | 0.69 | 0.60 | 0.64 | 0.88 | 0.71 | 0.61 | 0.76 | 0.78 |
| Person-Ground Contact (Rempe et al., 2021) | 0.71 | 0.68 | 0.68 | 0.71 | 0.69 | 0.70 | 0.73 | 0.68 | 0.74 | 0.70 | 0.73 | 0.72 | 0.73 |
| Foot-Floor Penetration (Rempe et al., 2021) | 0.70 | 0.70 | 0.69 | 0.70 | 0.69 | 0.70 | 0.71 | 0.70 | 0.69 | 0.70 | 0.69 | 0.69 | 0.69 |
| Physical Foot Contact (Tseng et al., 2023) | 0.69 | 0.69 | 0.69 | 0.69 | 0.69 | 0.69 | 0.69 | 0.69 | 0.69 | 0.69 | 0.69 | 0.69 | 0.69 |
| MoBERT (Voas et al., 2023) | 0.69 | 0.69 | 0.69 | 0.69 | 0.69 | 0.69 | 0.69 | 0.69 | 0.69 | 0.70 | 0.69 | 0.69 | 0.69 |
| MotionCritic (Ours) | $0.51_{\pm0.01}$ | $0.52_{\pm0.02}$ | $0.50_{\pm0.01}$ | $0.51_{\pm0.02}$ | $0.56_{\pm0.02}$ | $0.54_{\pm0.02}$ | $0.54_{\pm0.02}$ | $0.59_{\pm0.03}$ | $0.59_{\pm0.01}$ | $0.57_{\pm0.01}$ | $0.53_{\pm0.02}$ | $0.55_{\pm0.01}$ | $0.55_{\pm0.02}$ |

Table 6: Log-loss comparison of motion evaluation metrics on HumanAct12 action classes.

**Dataset Splitting and Model Retraining.** MotionPercept training and testing split is randomly divided in the paper. Here we also investigate into division based on annotator IDs. By adopting this approach, we ensure that each annotator's data is used exclusively in one of the sets, preventing overlap. Subsequently, we retrained and evaluated the model using this revised split. Disalignment between annotators as shown in Figure 2 results in slight decline of MotionCritic performance. Results are shown in the following table and align with our initial expectations.

| Metric | MDM | | FLAME | |
|---|---|---|---|---|
| | Acc. (%) ↑ | Log Loss ↓ | Acc. (%) ↑ | Log Loss ↓ |
| Root AVE | 59.22 | 0.6875 | 48.42 | 0.6984 |
| Root AE | 62.10 | 0.6787 | 59.54 | 0.6711 |
| Joint AVE | 57.01 | 0.6875 | 44.61 | 0.6973 |
| Joint AE | 63.17 | 0.6790 | 58.37 | 0.6891 |
| Jerk | 66.33 | 0.7476 | 65.84 | 0.5984 |
| Acceleration (Yang et al., 2023) | 63.82 | 0.7641 | 66.67 | 0.6919 |
| Person-Ground Contact (Rempe et al., 2021) | 68.08 | 0.6835 | 69.82 | 0.7243 |
| Foot-Floor Penetration (Rempe et al., 2021) | 51.77 | 0.6972 | 55.56 | 0.6906 |
| Physical Foot Contact (Tseng et al., 2023) | 66.52 | 0.6927 | 66.00 | 0.6930 |
| PoseNDF (Tiwari et al., 2022) | 56.00 | 0.6930 | 53.07 | 0.6931 |
| NPSS (Gopalakrishnan et al., 2019) | 53.48 | 0.6975 | 52.07 | 0.6944 |
| NDMS (Tanke et al., 2021) | 62.08 | 0.6590 | 63.35 | 0.6301 |
| MoBERT (Voas et al., 2023) | 47.49 | 0.6933 | 52.40 | 0.6932 |
| MotionCritic (Ours) | **80.13** | **0.5402** | **79.24** | **0.5927** |

Table 7: Quantitative comparison of metrics on re-split testsets of MotionPercept.

**Inplementation Details of Other Metrics**   Here we provide details of the implementation of the compared metrics. To ensure reproducibility, we will publish the code for all metrics. Overall, we utilized official code and models whenever possible, making only minimal and reasonable modifications to meet our evaluation needs.

- **Root AVE, Root AE, Joint AVE, and Joint AE:** These metrics were implemented following the definitions provided by Voas et al. (2023).

- **Acceleration and Jerk:** We used the released official implementations and hyperparameters from Yang et al. (2023) without any modifications. Jerk was derived as the first derivative of Acceleration.

- **Person-Ground Contact and Foot-Floor Penetration:** We used the released official implementations and hyperparameters from Rempe et al. (2021) without any modifications.

- **Physical Foot Contact:** We followed the approach described in Tseng et al. (2023), with slight adaptations to rewrite the original NumPy implementation into PyTorch.

- **PoseNDF:** For each motion sequence, we extracted all frames as poses and calculated neural distances using the official pretrained models provided by Tiwari et al. (2022). We implemented two methods: one using the average distances across all frames as the motion evaluation metric, and the other using the maximum distance among all frames. The latter performed better and was reported in Table 1.

- **NPSS:** We used the released official implementations and hyperparameters from Karras et al. (2019) without any modifications. Motions were matched to corresponding ground-truth sequences in the HumanACT12 action class, as NPSS is uni-modal and compares motions to single ground-truth sequences.

- **NDMS:** We used the released official implementations and hyperparameters from Tanke et al. (2021) without any modifications. We first transformed axis-angle SMPL into xyz coordinates and utilized code from official implementation for skeletons.

- **MoBERT:** We utilized their pretrained model from Voas et al. (2023) and computed the Naturalness metric, which evaluates the intrinsic plausibility of the motion itself and aligns with the theme of our work.

This setup ensures that our evaluation metrics are consistent with the official implementations and provides a clear methodology for reproducibility.

## B.2   FAILURE MODES

In addition, we analyze the failure modes of MotionCritic when it is used as a motion quality metric (Table 8). To do this, we computed the critic score differences for all samples in the test set (user-annotated better vs. user-annotated worse) and summarized the statistics in the table below:

| Metric | Mean | Std | Max | Min |
|---|---|---|---|---|
| All Scores | 4.32 | 3.72 | 29.82 | -11.73 |
| Correct Cases (85.1%) | 4.75 | 3.79 | 29.82 | 6.79e-5 |
| Wrong Cases (14.9%) | -1.84 | 1.91 | -1.91e-4 | -11.73 |

Table 8: Critic score statistics for all cases, correct cases, and wrong cases.

We note that in correct cases, the critic score differences are larger, indicating higher confidence. Conversely, in the wrong cases, the score differences are smaller, suggesting that the errors occur when the model is less confident. We also provide the distributions of critic scores for both correct and wrong predictions (see Fig. 10). On the left, the vertical axis represents the number of examples (with wrong cases comprising 14.9% and correct cases 85.1%). On the right, the vertical axis shows the percentage within the corresponding part.

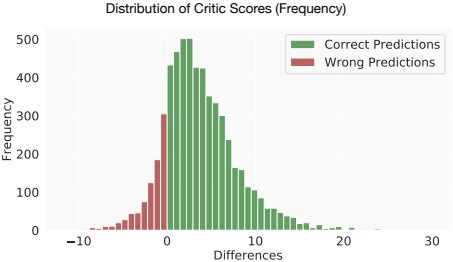 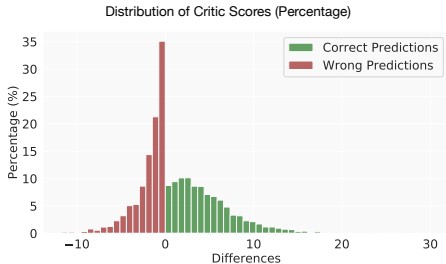

Figure 10: Distributions of critic scores' differences for correct and wrong predictions.

It is clear that in cases where the model predicts incorrectly, the majority of the score differences are very small. For example, 35.10% of the wrong cases have a critic score difference within 0.73 (one bin), whereas only 8.74% of correct cases fall into the same range. This indicates that many of the incorrect predictions occur when the critic scores are close, rather than due to large, obvious errors. In contrast, correct predictions tend to have larger score differences, reflecting higher confidence from the model. Additionally, some errors occur in particularly challenging examples where even human annotators struggle to differentiate, and different individuals may make varying judgments.

## B.3 SENSITIVITY ANALYSIS

We conducted a preliminary sensitivity analysis by perturbing AMASS ground-truth (GT) motions with Gaussian noise of various scales. Assuming that GT motions should consistently receive higher scores than perturbed ones, we evaluated the critic's ability to distinguish between them.

**Robustness.**  `MotionCritic` demonstrates robustness in ranking GT motions above perturbed ones across noise scales, as demonstrated in Figure 11(A). We also observed how the average and standard deviation of critic scores vary with different noise scales, as demonstrated in Figure 11(B).

**Smoothness.**  By analyzing the 3D relationship between noise scale, critic scores, and perturbed critic scores, we observed steep score changes near the natural motion distribution, as demonstrated in Figure 12(B). The interactive version of this 3D visualization is available at `https://iclrauthors6688.github.io/iclr6688/sensitivity_analysis_3d.html`, highlighting the critic's sensitivity outside the natural motion distribution. We also provide a 2D scatter plot focusing on perturbed-score vs noise-scale, as demonstrated in Figure 12(A).

**Non-smooth energy landscape.**  In summary, we discover that the critic model exhibits a non-smooth and "bumpy" energy landscape, which serves as a double-edged sword in our framework. On one hand, this non-smoothness is a desirable property, making the critic highly discriminative and sensitive to subtle motion perturbations, aligning well with human perceptual abilities. On the other hand, it introduces challenges to motion optimization. Future work could investigate better optimization methods, such as leveraging reinforcement learning strategies (Liu et al., 2024), to mitigate challenges associated with this non-smoothness.

**Non-distance measure.**  Based on the training objective, the Bradley-Terry model, the critic scores are intended to represent relative preferences rather than absolute distances. Specifically, the scores indicate whether one motion is judged as superior to another, rather than their distance. Therefore, it's worth noticing that the critic score should not be interpreted as a distance function. This design aligns with the goal of producing a ranking-based evaluation metric rather than a continuous distance measure.

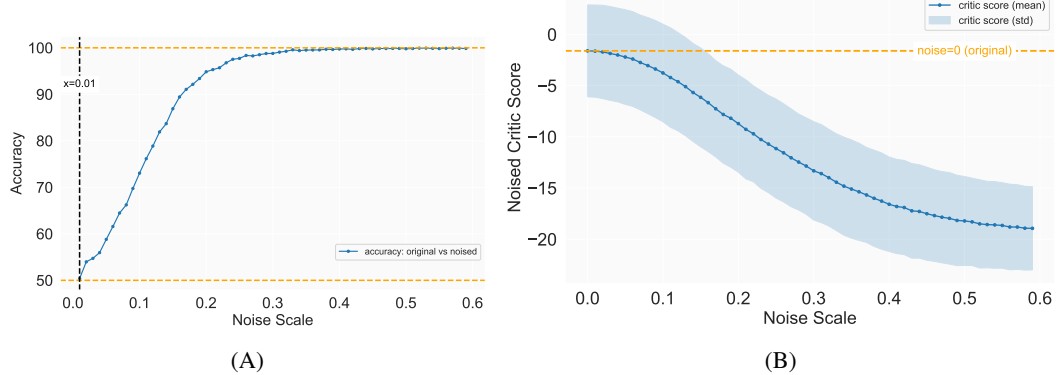

Figure 11: Sensitivity analysis results. (A) Accuracy vs noise-scale curve. (B) Average and standard deviation of critic scores vs noise-scale.

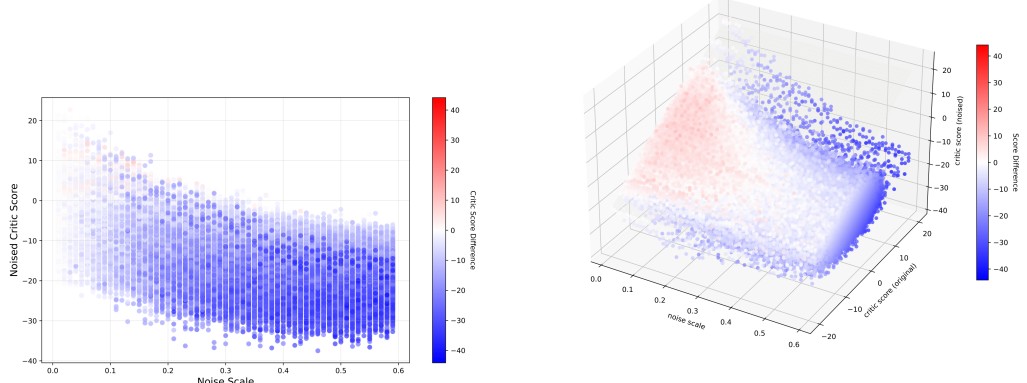

(A): 2D scatter plot of perturbed-score vs noise-scale.

(B): 3D scatter plot of noise scale, critic scores, and perturbed critic scores.

Figure 12: Sensitivity analysis upon AMASS GT motions. Critic score differences (original - perturbed) are demonstrated by colors.

## B.4 TEST ON OUT-OF-DISTRIBUTION MOTIONS

We test our method on out-of-diftribution motion dataset apart from HumanACT12 and UESTC. Specifically, we conduct additional analyses on AMASS using MotionCritic and observed some interesting findings:

| Dataset | Mean | Std | Max | Min |
|---|---|---|---|---|
| AMASS | -1.10 | 4.14 | 20.77 | -18.99 |
| HumanAct12 | -2.22 | 4.68 | 12.14 | -21.96 |
| MDM | -2.08 | 4.65 | 12.83 | -30.65 |

Table 9: Critic score statistics of the two datasets.

- The critic scores on AMASS are generally higher than those on HumanAct12, which aligns with our expectations, while the critic score statistics of the two datasets are shown in Table 9.
- The actions with the lowest scores on AMASS were indeed outliers, often violating physical constraints such as gravity (mostly due to object interactions that appear unreasonable when

considering only the motion). At the same time, it was also able to assign relatively high scores to some previously unseen complex motions.

Current results show that AMASS has higher average scores, maximum scores, and minimum scores compared to MDM-generated motions, even though some outliers in AMASS significantly lowered the scores, as discussed above. This suggests that normal motions in AMASS generally outperform those generated by MDM. Additionally, we conduct a pairwise comparison between MDM-generated motions and HumanAct12 GT motions, testing both the critic scores and user study results, which are presented in the table below. HumanAct12 indeed has quality issues , and both the critic score and users often favored the MDM-generated motions. Furthermore, the critic score continues to align well with human users' judgments when comparing these two types of motions, demonstrating its effectiveness.

Table 10: User study on HumanAct12 GT and MDM-generated motions

| Motion Category (12 categories, 10 motion pairs each) | HumaAct12 Preferred | MDM Preferred | User Similar | Humact12 Critic | MDM Critic | User Disagreement |
|---|---|---|---|---|---|---|
| 0 | 10% | 70% | 20% | 20% | 80% | 20% |
| 1 | 30% | 40% | 30% | 30% | 70% | 20% |
| 2 | 30% | 50% | 20% | 20% | 80% | 20% |
| 3 | 20% | 50% | 30% | 20% | 80% | 10% |
| 4 | 0% | 80% | 20% | 10% | 90% | 0% |
| 5 | 0% | 90% | 10% | 0% | 100% | 0% |
| 6 | 40% | 10% | 50% | 70% | 30% | 0% |
| 7 | 0% | 90% | 10% | 10% | 90% | 0% |
| 8 | 10% | 90% | 0% | 40% | 60% | 30% |
| 9 | 10% | 80% | 10% | 50% | 50% | 30% |
| 10 | 0% | 60% | 40% | 10% | 90% | 0% |
| 11 | 40% | 40% | 20% | 30% | 70% | 20% |
| **Overall (120 pairs)** | 15.83% | 62.50% | 21.67% | 25.83% | 74.17% | 12.50% |

1. **HumanAct12 User Preference:** This column shows the percentage of cases where users preferred the HumanAct12 motions over MDM.

2. **MDM User Preference:** This column shows the percentage of cases where users preferred the MDM motions over the HumanAct12 motions.

3. **User Similar:** This column indicates the percentage of cases where users found it difficult to distinguish between the HumanAct12 and MDM motions.

4. **HumanAct12 Critic Score Win:** This shows the percentage of cases where the HumanAct12 motion received a higher critic score.

5. **MDM Critic Score Win:** This shows the percentage of cases where the MDM motion received a higher critic score.

6. **User Disagreement with Critic Score:** This column indicates the percentage of cases where the user's choice did not align with the critic score.

These examples demonstrate that MotionCritic is an effective tool for automatically comparing and evaluating motion quality, whether individually or in bulk. It also helps uncover valuable insights, further highlighting its utility and potential.

## B.5 IMPACT OF SEQUENCE LENGTH, FRAMERATE, AND INTERPOLATION

To further address concerns regarding the impact of sequence length, framerate, and interpolation on the performance of the MotionCritic model, we conduct a detailed analysis using the AMASS GT dataset.

**Dataset and Pre-processing.** We extracted a total of 6,346 OOD (Out-of-Distribution) motion examples from the AMASS GT dataset. Among these, 547 examples (approximately 8.62%) have lengths shorter than 60 frames. We analyzed longer and shorter OOD motions separately to evaluate the model's performance under varying conditions. To further assess the robustness of the MotionCritic model, we generated a pseudo-GT dataset by adding Gaussian noise at different scales to the motions. We formed better-worse pairs by comparing unnoised and noised versions of the same motion.

For longer motions, we experimented with three different pre-processing approaches:

1. **Unprocessed Motion:** The full-length motion was fed directly into the model to obtain critic scores.
2. **Cut-down to 60 Frames:** The motion was truncated to 60 frames before being input into the model, expanding the dataset to a total of 12,011 motions.
3. **Uniform Frame Extraction:** 60 frames were uniformly extracted from the complete motion, which was then fed into the model for critic scores.

The results, presented as shown in Figure 13, demonstrate that the Critic model performs optimally when the motion lengths are 60 frames, consistent with the original dataset.

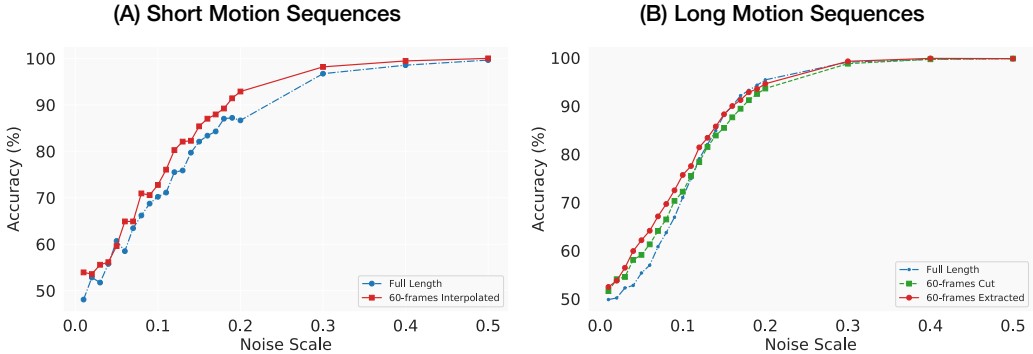

Figure 13: Performance of the Critic model across different motion lengths and interpolations.

For shorter motions, two pre-processing approaches were evaluated:

1. **Unprocessed Motion:** The full-length, short motion was input directly into the model for critic scores.
2. **Interpolation to 60 Frames:** The motion was uniformly interpolated to 60 frames before being input into the model.

The results, also presented as accuracy-noise line plots (on the left, Fig (A)), indicate that the Critic model can effectively handle shorter sequences with minimal performance loss after interpolation.

The results demonstrate that `MotionCritic` works best when the motion sequences are 60 frames in length, consistent with the training dataset, which could be achieved through interpolation or extraction. The model also exhibits strong performance across variable lengths and framerates, with only minor reductions in accuracy. This robustness highlights the advantages of the MotionCritic model.

## B.6 ENSEMBLE LEARNING USING DIFFERENT METRICS

To further explore the interrelationship between the different motion quality metrics, we treat the different metrics as features and train a simple classifier to perform binary classification (ensemble learning). To train the ensemble model, we create a separate train-test split from the `MotionPercept` MDM test set. We explore three ensemble approaches to combine the metrics and evaluate the complementary effects between the heuristics and `MotionCritic`:

| Metric | Accuracy (%) |
|---|---|
| `MotionCritic` | **85.07** |
| Acceleration | 64.26 |
| Jerk | 65.48 |
| GroundContact | 71.78 |
| PFC | 64.79 |

Table 11: Initial accuracy of metrics on the MotionPercept MDM test set.

| Method | Accuracy (%) |
|---|---|
| Logistic Regression | 81.12 |
| SVM | 82.49 |
| MLP | **85.17** |
| `MotionCritic` Only | 84.96 |

Table 12: Accuracy of ensemble learning methods.

1. **Logistic Regression.** Binary classification using the five normalized features.

2. **Support Vector Machine (SVM).** Binary classification using the five features across two motions in a better-worse pair.

3. **Multilayer Perceptron (MLP).** A simple MLP with one hidden layer of 128 dimensions was trained using the five features as input, producing a single score to determine better or worse outcomes. The training objective was based on the Bradley-Terry model, similar to MotionCritic.

The accuracy of each combined method is presented in Table 12. The results indicate that combining the metrics does not significantly improve accuracy beyond what MotionCritic alone can achieve, suggesting that MotionCritic already incorporates much of the heuristic knowledge.

**Visualization of Heuristics and MotionCritic Overlap**    We visualized the overlap between heuristics and MotionCritic on the entire MotionPercept MDM test set using a Venn diagram at Figure 14, demonstrating a significant overlap in correctly classified pairs. This suggests that when MotionCritic makes a mistake, other metrics tend to err in the same cases.

**Ablation Study in SVM**    To further understand the contribution of each metric, we conducted an ablation study using SVM by incrementally adding features in the order of [`MotionCritic`, Acceleration, Jerk, GroundContact, PFC]. The results are shown in Table 13.

| Features Added | Accuracy (%) |
|---|---|
| `MotionCritic` | 82.49 |
| `MotionCritic` + Acceleration | 82.40 |
| `MotionCritic` + Acceleration + Jerk | 82.45 |
| `MotionCritic` + Acceleration + Jerk + GroundContact | 82.49 |
| All Features | 82.49 |

Table 13: Ablation study results in SVM.

**Feature Removal in SVM**    We also conducted experiments by sequentially removing features to observe the impact on accuracy, as presented in Table 14. The results indicate that removing MotionCritic has a substantial negative impact on model performance, while removing other features has a minimal effect.

**Logistic Regression Coefficients**    The coefficients from the logistic regression model are presented in Table 15. The positive coefficient for MotionCritic indicates that higher values are associated with better outcomes, while the negative coefficients for other metrics suggest that lower values are preferable.

**SHAP Values Visualization**    We visualized the SHAP (SHapley Additive exPlanations) values, which provide insights into the contribution of each feature to the model's predictions (see Figure 14). The figures show the SHAP values for the five metrics tested on a series of samples from our test set, evaluated with the MLP ensemble model. Across these samples, the SHAP values for the

| Feature Removed | Accuracy (%) |
|---|---|
| MotionCritic | 66.52 |
| Acceleration | 82.49 |
| Jerk | 82.40 |
| GroundContact | 82.40 |
| PFC | 82.49 |

Table 14: Accuracy after feature removal in SVM.

| Feature | Coefficient |
|---|---|
| MotionCritic | 2.49 |
| Acceleration | -0.61 |
| Jerk | -0.33 |
| GroundContact | -0.39 |
| PFC | -0.07 |

Table 15: Logistic regression coefficients.

MotionCritic feature generally have the largest absolute values, indicating that the critic score is the most important feature. Additionally, when the critic score is high, its SHAP value is positive, suggesting a positive correlation between the critic score and motion quality, which aligns with expectations.

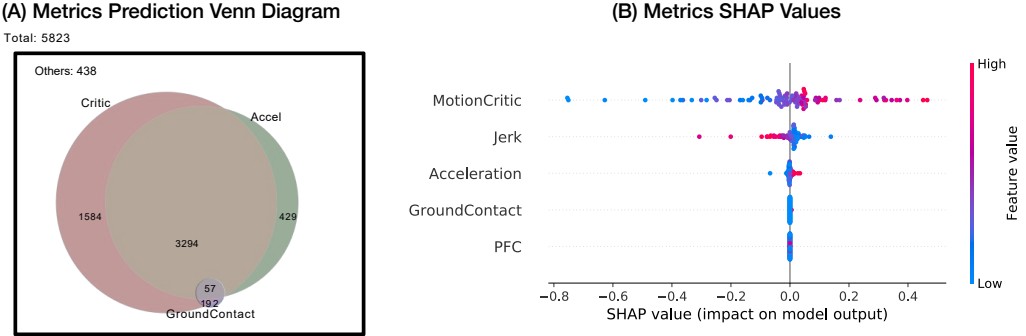

Figure 14: Metrics' venn fiagram and SHAP values.

The experimental results demonstrate that MotionCritic effectively captures the essential knowledge provided by the heuristics. Attempts to further enhance accuracy by combining MotionCritic with other heuristics yield marginal improvements, underscoring the robustness of MotionCritic in this context.

## C  DETAILS ON MOTIONCRITIC: AS TRAINING SUPERVISION

### C.1  FINE-TUNING

**Critic Score Clipping.**    Generally, a higher `MotionCritic` score indicates better motion quality. However, this relationship has an upper limit. During our fine-tuning process, we clip motions with reward scores exceeding a threshold $\tau$ when computing gradients before back-propagation. This threshold, determined through a series of comparative experiments, is set at $\tau = 12.0$, approximately the upper bound of ground-truth critic scores. We found that this setting yields the best results. Fine-tuned motion generation models without reward clipping tend to artificially inflate reward scores on a few specific motions, which increases the average `MotionCritic` score but degrades overall performance. Thus, reward clipping is essential to maintain the integrity and quality of the fine-tuning process.

**Finetuning Details.**    Inspired by ImageReward(Xu et al., 2024), we observe how the critic score changes over denoising steps to identify the optimal time window for ReFL intercept. As shown in Figure 15(A), we set the hyperparameter step sampling range to $[T_1, T_2] = [700, 900]$, where the critic score witnesses a rapid increase. The full fine-tuning curve, provided in Figure 15(C), demonstrates that the critic output improves rapidly during the early stages of fine-tuning and stabilizes after a certain point.

In addition, another discussion by Clark et al. (2024) proposed DRaFT-LV, employing single-step gradient backpropagation, observed to be the most efficient. To evaluate the impact of alternative step sampling strategies, we conducted experiments comparing our current setup with DRaFT-LV's methodology. The results, detailed in Figure 15(C), show that single-step refinement can indeed improve fine-tuning efficiency, consistent with existing findings.

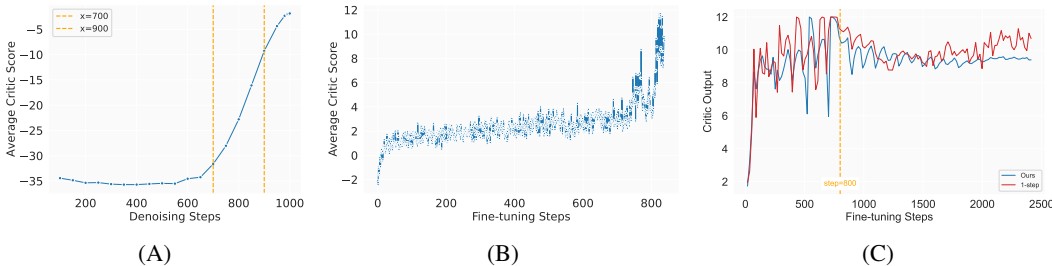

|  |  |  |
|:---:|:---:|:---:|
| (A) | (B) | (C) |

Figure 15: Fine-tuning process. (A): Critic score in 1000-step denoising process. (B): Critic output in 800-step fine-tuning process. (C): Full finetuning process of our strategy based on ReFL (Xu et al., 2024) and 1-step back-propagation based on DRaFT-LV (Clark et al., 2024).

### C.2  FINETUNED RESULTS

**Improved Critic Score.**    As shown in Figure 16, the critic score increases after `MotionCritic` supervised fine-tuning. This scatter plot collects all data points from the test set, with the critic score of motions before fine-tuning on the x-axis and the critic score of the corresponding motions after fine-tuning on the y-axis. As demonstrated in Figure 16(A), we first compare results with and without critic model supervision. In the latter case, the original MDM loss is used for continued training without our `MotionCritic`-based plug-and-play module. The scatter plot clearly indicates that the results with critic model supervised fine-tuning achieve significantly higher scores. The second experiment in Figure 16(B) examines different fine-tuning steps using 800 steps from the first set as a baseline. The results demonstrate that critic model supervised fine-tuning consistently improves the critic score throughout the fine-tuning process.

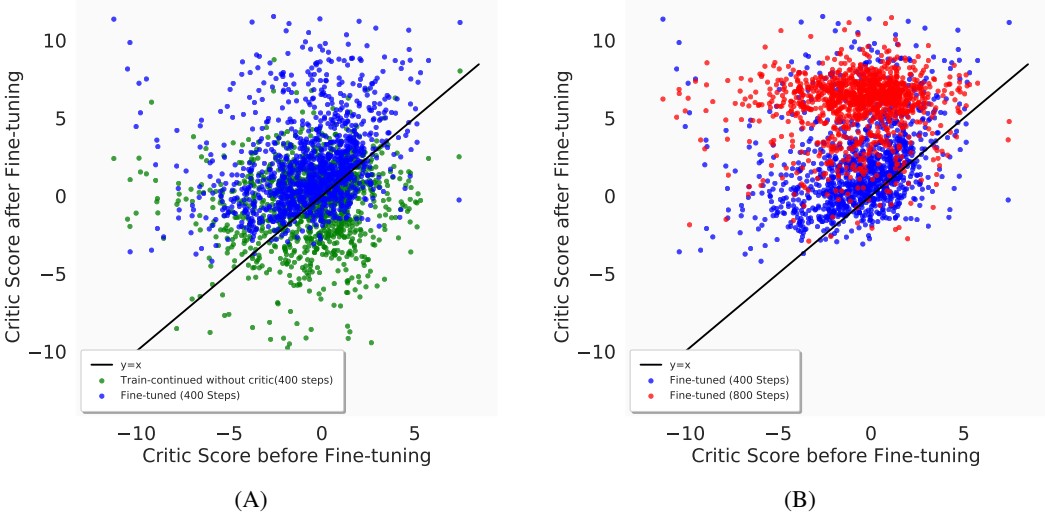

Figure 16: Visualization of critic scores on fine-tuning experiments. (A): Fine-tuning 400 steps with and without `MotionCritic` supervision compared. (B): Fine-tuning with 400 and 800 steps compared.

**Improved Motion Quality.**  We conduct an independent user study to compare motion pairs generated at various fine-tuning stages and calculate the Elo Rating (Elo, 1978; Tseng et al., 2023). Figure 17 demonstrates that the quality of motions consistently enhance as fine-tuning advances, as indicated by the user study. This improvement aligns with the training objective of elevating the critic score.

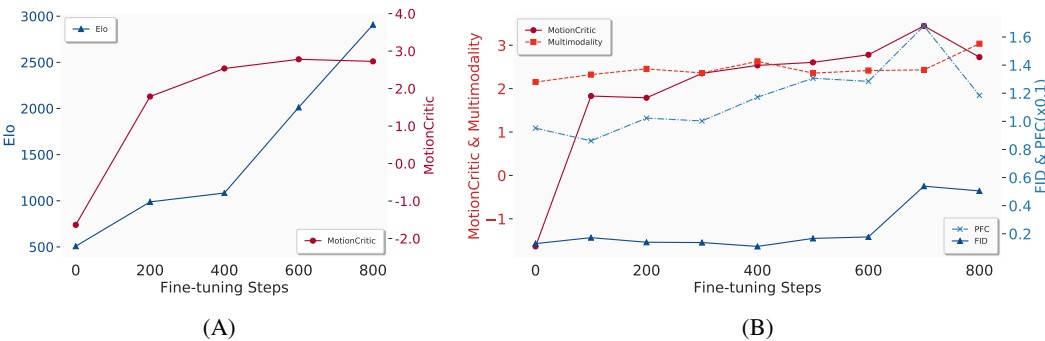

Figure 17: Results from fine-tuning process. (A): Elo ratings and Critic scores. (B): FID, PFC(Tseng et al., 2023), Multimodality and Critic scores.

We further inspect the change of different metrics during the fine-tuning process in Figure 17(B) and Table 2. PFC (Tseng et al., 2023) and FID are expected to be negatively correlated with motion quality (the smaller, the better), and `MotionCritic` and multimodality are expected to be positively correlated (the greater, the better). The results indicate that existing motion quality metrics (*e.g.* FID, PFC) do not adequately reflect human preference, as they poorly correlate with Elo ratings from user studies. Meanwhile, improving the critic score does not necessarily conflict with the multimodality metric, which models the diversity of generated motions.

# D  DETAILS ON USER STUDIES

## D.1  ANNOTATION DETAILS

The annotators of `MotionPercept` are recruited by an annotation company and are ordinary adults (5 men and 5 women). Previous psychological research has shown that the ability to perceive and discern biological motion is universal and consistent, with a neural basis  (Johansson, 1973), and does not rely on specific training or professional background. This finding is also consistent with what we observed in Figure 2. As for potential biases, while some degree of annotator bias may exist (as with any human-annotated dataset), we believe it is minor and manageable, as indicated by our analysis in Section 3. Overall, we consider the opinions of the annotators to be representative.

## D.2  STAND-ALONE USER STUDIES

**Details.**    We conduct user studies on GT subsets grouped from HumanAct12 (Guo et al., 2020) and motions generated during finetune steps as discussed in the main text. Our user study platform is shown in Fig 18. In user study, one motion pair of two motions are played simultaneously, with their critic scores and text prompts being hidden. Annotators should choose the better motion or choose "Almost the Same" if they can't make a decision. We perform user study on 5 different finetune steps and 5 GT batches grouped from HumanAct12 (Guo et al., 2020).

**Different annotators.**    For the stand-alone user study, we did not use large-scale annotators from the annotation company. Instead, we recruit new volunteers through a campus announcement who performed pairwise comparisons of two motions, selecting the better one. The five volunteers are students from two different universities with backgrounds in computer science, engineering, arts, and history. During this process, they do not see any additional information such as scores or text. We ensure that they receive reasonable compensation for their time and effort. We then calculate the Elo Rating based on their results. The annotation interface and rating formulas can be found in this section as well.

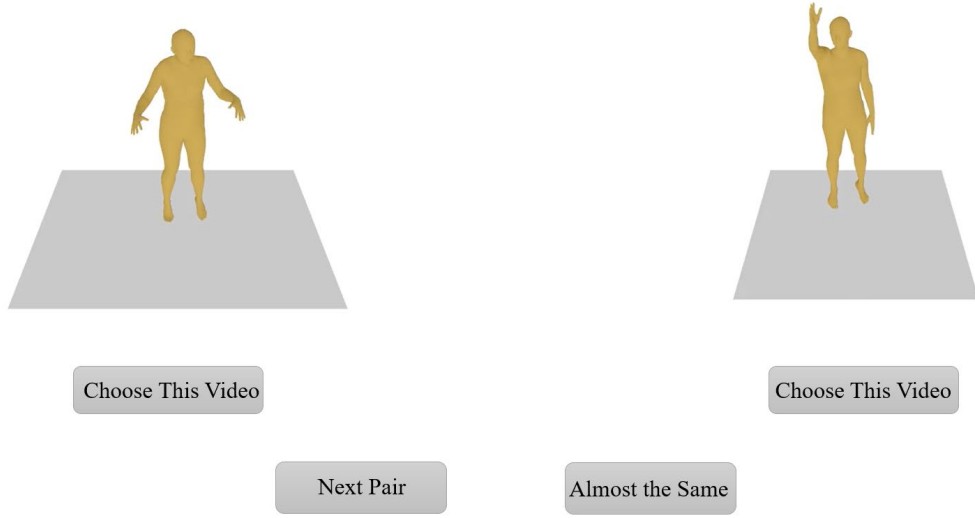

Figure 18: The interface of our user study platform.

### D.3 RATING FORMULAS

**Win-rates.** After annotation, we calculate win-rates of subsets pairs. In user study, each subset has the same amount of motions. Given subsets pair $(A, B)$, win-rates shows the percentage of motion pairs where motion of subset $A$ win over motion of subset $B$ in naturalness. Then we paint heatmaps of all subsets with their win rates. Since the result of one match maybe tie, the sum of win-rates of two subsets in a pair and data in symmetric positions of heatmap might be less than 1.

**Elo Rating**    (Elo, 1978; Tseng et al., 2023) After annotation, we calculate elo rating of each subsets as follows:
Suggest $R_A, R_B$ are the initial ratings of two compared subsets $A$ and $B$. The expected win rate of subset9s $A$ and $B$, denoting as $E_A, E_B$ can be calculated as follows:

$$E_A = \frac{1}{1 + 10^{(R_B - R_A)/400}} \tag{7}$$

$$E_B = \frac{1}{1 + 10^{(R_A - R_B)/400}} \tag{8}$$

The new ratings of subsets $A$ and $B$ are:

$$R'_A = R_A + K(S_A - E_A) \tag{9}$$

$$R'_B = R_B + K(S_B - E_B) \tag{10}$$

where K is rating coefficient, we choose 32; and $S$ is real score, which is 1 for winner, 0 for loser and 0.5 if the result is a tie. We set the initial rating of each subset as 1500.

