# OpenReview forum: "Aligning Human Motion Generation with Human Perceptions"
_ICLR.cc/2025/Conference — ICLR 2025 Poster_

### Official Review · Reviewer_pzfu · 2024-11-01

**Soundness:** 2
**Presentation:** 2
**Contribution:** 2
**Rating:** 6
**Confidence:** 4

**Summary:**

This paper addresses the challenges in evaluating and improving human motion generation by proposing a human-aligned evaluation approach. The authors introduce MotionPercept, a large-scale dataset containing over 52,000 human preference annotations for generated motions, which provides a foundation for understanding motion quality from a human perspective. They also develop MotionCritic, a model trained on MotionPercept, which accurately predicts human-perceived motion quality and surpasses traditional metrics in alignment with human preferences. MotionCritic can be easily integrated into existing generation pipelines, where it functions as a supervision signal to improve motion quality with minimal fine-tuning.

**Strengths:**

A large amount of human preference annotation work has been done for text-to-motion, and a sizable dataset has been provided, which is beneficial to the community.

**Weaknesses:**

The human preference annotation lacks consideration for the diversity of motion.

**Questions:**

1. There is inherent diversity in the motion generated from text-to-motion. Regarding the fine-tuning comparisons in the supplementary materials, I observe that the generated motions generally align with semantic accuracy, and the physical accuracy appears to improve over time. Could it be that the critic score primarily reflects physical correctness? What I’d like to see is a visualization that compares multiple generated motions for a single text prompt, showing the distribution of critic scores (with five or more samples), as well as how motion diversity changes before and after fine-tuning. This result would better demonstrate the effectiveness of your proposed method and its balance in generating diverse outputs.

2. When the text is simple, such as "run" or "punching," the corresponding motions should naturally be diverse. However, based on the annotation process provided in the HTML document, it seems that only the best out of four generated motions is selected, which shows no particular consideration for diversity. This raises the question of how Multimodality would improve after fine-tuning. [1] aligns motion generation with human preference. They only select the best result, which significantly reduces diversity (MModality, the diversity of motion generated for one text description). A discussion with [1] would also be beneficial.

3. The annotation of human preference in this paper for motion follows the same process as for images. I see that many useful insights (as metrics and supervision) also resemble those used in image-based tasks. Since motion data is a unique form of 3D data, differing greatly from 2D image data, are there any distinct insights here?

4. The annotation of human preference in this paper for motion follows the same process as for images. I also see that many insights (as metrics and supervision) also the same as image-based tasks [2,3]. Since motion data is a unique form of 3D data, differing greatly from 2D image data, are there any distinct insights here?

5. In Table 1, where MDM’s generated results are evaluated with different metrics, why does the metric with higher Acc is better?

Reference:

[1] HuTuMotion: Human-Tuned Latent Motion Diffusion Models with Minimal Feedback.
[2] ImageReward: Learning and Evaluating Human Preferences for Text-to-Image Generation.
[3] Better Aligning Text-to-Image Models with Human Preference.

---

> ### Comment · Reviewer_pzfu · 2024-11-26
>
> Thank you to the author for the rebuttal, but I still feel that the overall approach is not compelling enough, and there are logical issues in the explanation.
>
> 1. Your feedback annotation is based on text-to-motion, which certainly involves text alignment factors. Even if, as you mentioned, using human preferences to emphasize physicality for improving generative model performance seems plausible, it is not entirely convincing. Physicality should instead focus on physical simulations and joint motion, as joint motion data can directly calculate physical performance.
>
> 2. Thank you for providing the extra visualizations, but upon closer inspection, the results are not satisfactory. For instance, some high-scoring samples still show obvious jittering and sliding steps, while some lower-scoring samples exhibit fewer issues.
>
> 3.''Utilizes a critic model to fine-tune the motion generation model in a more flexible, unconstrained manner, which does not hurt multimodality '' is not convincing. The critic model is essentially just a discriminator. This statement does not provide any meaningful insight.

---

> ### Author Response · Authors · 2024-11-26
> **Rebuttal by Authors (1/2)**
>
> We thank reviewer **pzfu** for the thoughtful feedback. We are pleased to know that our rebuttal addressed most of your initial concerns, such as the **diversity of generation results** and the rationale for **why higher accuracy is better in Table 1**. However, we are confused by the fact that the score was changed from **5 to 3** within two minutes of the comment saying, *“I will maintain my score.”* Was this a misunderstanding or a misclick? We would greatly appreciate clarification.
>
> Now, we will address your new concerns point by point.
>
> **Q1**: Text alignment factors
>
> **A1:** Evaluation of motion generation typically involves multiple dimensions, such as naturalness/fidelity, diversity, multi-modality, and alignment with text/audio/scene. In this work, we specifically focus on improving one critical aspect of motion generation: **motion quality** (e.g., naturalness, smoothness, plausibility), which we align with human perception. As explained in Appendix A.3, we deliberately exclude the text descriptions of the motion during the annotation process. This is because our primary objective is to evaluate the quality of the motion itself, independently of its alignment with textual descriptions.
>
> **Q2**: Physicality and joint motion data
>
> **A2:** While **physicality** is an important aspect of motion quality, it is not equivalent to the **human perception of motion quality**, which includes a broader range of factors.
>
> - **Conceptually:** Our study goes beyond **objective physical and biomechanical plausibility** to include subjective factors like **naturalness** and **smoothness**, which are strongly influenced by human perception and difficult to quantify using purely physical metrics. This is why we conducted a **large-scale human perceptual study** to learn these standards in a **data-driven manner**.
>
>     For example, a motion that satisfies physical constraints (e.g., no joint violations or unnatural forces) may still feel unnatural to humans—such as a rapid, jerky movement that, while physically plausible, appears awkward and unrealistic.
>
> - **Experimentally:** As demonstrated in Section 5.2, our proposed MotionCritic significantly outperforms prior handcrafted physical metrics (e.g., Person-Ground Contact, Foot-Floor Penetration, Physical Foot Contact) in reflecting human preferences. Additionally, as detailed in Appendix B.5, MotionCritic implicitly incorporates information from these physical metrics through its data-driven framework.
>
> Nonetheless, We agree that combining physical simulation priors with our model's human perceptual priors is a meaningful direction for future exploration. By integrating these two complementary aspects, we can potentially develop models that are both physically grounded and perceptually realistic, leading to more natural and convincing motion evaluation and synthesis.
>
> **Q3**: Extra visualization results
>
> **A3**: As per your earlier suggestion, we visualized the motion both before and after fine-tuning, which is intended to demonstrate that our fine-tuning process does not negatively impact generation diversity. It is worth noting that these results are **completely free of cherry-picking**.
>
> Regarding your observation that some lower-scoring samples may appear better than some higher-scoring ones—this is entirely reasonable. As reported in Table 1, our metric’s accuracy is not 100%, so occasional failure cases are expected. However, statistically, our proposed metric demonstrates significantly better alignment with human preferences compared to all existing metrics, as shown by the comprehensive evaluations.
>
> We also examined these failure cases in Appendix B.2 and found that they are relatively rare. In such cases, the differences in Critic scores tend to be smaller, and the motions are generally more ambiguous and harder to judge. This reflects the inherent subjectivity in human perception, where different individuals may interpret the same motion differently, especially for borderline cases. Thank you again for your constructive feedback, and we hope this explanation clarifies the intent and outcomes of our visualizations.

---

> > ### Comment · Reviewer_pzfu · 2024-11-27
> >
> > Because the author's previous response had some logical issues, I reduced my score from 5 to 3. However, now that some concerns have been addressed, I am raising my score to 6. I can understand how motion preference annotations can improve the model. However, overall, motion is a video, and the introduction of the time dimension leads to too much freedom. Moreover, the data-driven preference annotations for naturalness, physicality, etc., require an excessive amount of annotations, which is not an efficient approach. This remains a limitation of the paper.

---

> > > ### Author Response · Authors · 2024-11-27
> > > **Thanks for your feedback**
> > >
> > > Thank you for letting us know that your concern regarding diversity has been addressed, and for updating the score to 6. We appreciate your feedback, and in a future revision, we will incorporate the discussion from the rebuttal, including the limitations of our method.

---

> ### Author Response · Authors · 2024-11-26
> **Rebuttal by Authors (2/2)**
>
> **Q4**: “Utilizes a critic model to fine-tune the motion generation model in a more flexible, unconstrained manner, which does not hurt multimodality” is not convincing.
>
> **A4**: The statement was made in comparison to the work you referenced, HuTuMotion [1], and we would like to expand on this discussion below:
>
> - **Flexibility**: In HuTuMotion [1], the base motion generation diffusion model remains unchanged, as their approach does not fine-tune the motion generation model itself. In contrast, our method fine-tunes the motion generation diffusion model directly, making our approach more flexible.
> - **Unconstrained**: HuTuMotion [1] constructs an optimal prior distribution for texts that are near a given representative text (RT). In their implementation, they use ChatGPT to generate a predefined set of RTs (typically around 5–50). This predefined set inherently constrains the diversity of the generated results (multimodality). In contrast, our method does not impose such presets, making it unconstrained in terms of generation.
>
> We acknowledge that HuTuMotion [1] has its own advantages, such as providing a minimal feedback-based approach, as we have mentioned earlier. However, based on the theoretical analysis above, our approach is more flexible and unconstrained, which explains why it does not reduce multimodality. This claim is further supported by our **experimental results** (see Table 2).
>
> Additionally, your earlier statement, "*They only select the best result, which significantly reduces diversity*", is not entirely accurate. Selecting the "best" result based on human perception does not inherently conflict with motion diversity. High-quality motion inherently has multiple forms of expression, does it not? Our method does not assume that generating high-quality motion leads to a reduction in diversity, and this is further validated by our experimental results.

---

> > ### Comment · Reviewer_pzfu · 2024-11-27
> >
> > After my careful review, I found that the diversity in this paper is primarily maintained by the presence of many similar texts from a data-driven perspective. So, even though each text has only one preference annotation, as long as there are many annotations, similar texts correspond to a variety of human-preferred motions. This concern has now been addressed, although the author did not provide the reasonable explanation.

---

### Official Review · Reviewer_zYMP · 2024-11-03

**Soundness:** 3
**Presentation:** 4
**Contribution:** 3
**Rating:** 8
**Confidence:** 4

**Summary:**

The paper introduces a novel evaluation metric for 3D human motion that aligns more closely with high-level human perception. A large-scale data annotation effort is conducted by asking users to select the highest quality motion from a set of samples. To capture human preferences, the paper proposes training a critic model that learns to assign a quality score to each motion sample. A comprehensive analysis compares the proposed critic model with commonly used evaluation metrics from prior work, revealing deficiencies in these established metrics for aligning with human perception, while the proposed critic model shows much higher alignment. Additionally, the paper presents a fine-tuning approach for improving the motion quality of pre-trained motion diffusion models by leveraging the proposed critic model.

**Strengths:**

The paper addresses a fundamental problem in human motion generation and is both well-written and well-motivated. I appreciate the authors' meticulous effort in their work.

Annotating a large sample set and training a critic model on this data is a clever idea. Additionally, the thorough analysis and comparisons effectively demonstrate the proposed method's impact. While the human motion modeling field has converged on a set of complementary metrics for quality assessment, these metrics still fall short of aligning with human perception, as shown in this paper. The proposed MotionCritic takes a more direct approach to addressing this issue. I believe this work is highly valuable for the community, contributing both to the research on evaluation and analysis of future human motion synthesis research. I also hope it will serve as an alternative to the counterintuitive FID score.

**Weaknesses:**

A sensitivity analysis could be included to assess the robustness and smoothness (e.g., Lipschitz continuity) of the critic model. For instance, how would the critic model respond if a pose is slightly perturbed, either randomly or in a structured way by rotating a joint beyond its limits?

By design, certain artifacts—such as foot-floor penetration, person-ground contact, etc.—might be missed, as the critic model lacks information needed to detect these issues (e.g., it only uses rotations and root translation as inputs). What are the authors' thoughts on this limitation?

Another potential concern is the reliance on a single model (e.g., MDM) for generating data used in the annotations. Although the paper presents results with an additional model (e.g., FLAME) to demonstrate generalization, it may be beneficial to curate an annotation set from various models and architectures to mitigate possible biases.

**Questions:**

1- This aligns with my comment on sensitivity analysis. The paper addresses only fine-tuning but does not consider an end-to-end training scenario. Theoretically, the critic model could also be used to train a motion model from scratch. I mention this because, even in the MDM fine-tuning experiment, the critic term is tightly controlled (with a low learning rate, low critic weight, and high KL-term weight). Is this due to difficulties in training with the critic term, possibly because of a non-smooth energy landscape?

2- Locomotion sequences with significant foot sliding receive high critic scores. Could this be due to a bias in the annotated samples, given that MDM generates many sequences with foot sliding? Have the authors considered training the critic with ground-truth samples as well? This should be straightforward, as ground-truth samples could always be preferred over synthetic ones.

3- Are the critic scores for two different samples comparable? I assume it isn’t a distance function. Could the authors comment on this?

4- This is minor but it would be great to have a comparison against a log-likelihood metric such as a normalizing flow.

---

### Official Review · Reviewer_fHeh · 2024-11-03

**Soundness:** 2
**Presentation:** 3
**Contribution:** 3
**Rating:** 8
**Confidence:** 2

**Summary:**

This work introduces MotionPercept, a large-scale synthetic human motion dataset which contains human-ranked synthetic human motion generations, and MotionCritic, a model learned on MotionPercept. MotionCritic can be utilized to evaluate motion quality but also as an additional loss for motion quality.
The authors show that MotionCritic follows human perception closer than existing evaluation metrics and can even be used to find outliers in existing human motion datasets.

**Strengths:**

This paper addresses a common problem in generative motion synthesis: how to evaluate the motion quality of a generated sequence? In contrast to the often utilized FID score MotionCrititic operates on a per-sequence base, which allows for a much more granular evaluation.

**Weaknesses:**

One major concern of the motion critic is its quality ceiling: MotionPercept is highly dependent on the generative models that produces the motion, so in turn MotionCritic is highly dependent on those generative models as well. That means that, for the model, the best possible motion is the best generative model of MotionPerecpt, potentially limiting the usefulness of the critic once generative models produce significantly better motion than has been produced for MotionPercept. Could the authors comment if there is a way to remedy this or if they have considered this a problem?

Another concern would be: how does MotionCritic behave when provided with unseen motion modalities? I.e. lets say MotionCritic was not trained with dancing motion but is now tasked to judge the motion quality of dancing: do the authors expect MotionCritic to continue being reliable or would retraining be required?

Why did the authors choose 2.3s sequence lengths (60 frames @24Hz)? This seems very short for complex motion, i.e. a person “sitting down”. Were complex motions discarded for the dataset generation? Otherwise, only the motion initiation would have been recorded.
Did the authors observe that the model tends to start from the same pose or position, i.e. a T-Pose? This might limit the application of MotionCiritic as it could favor motions and poses that are more similar to the “initialization” pose/motion.

How can MotionCritic be applied to sequences of different lengths, i.e. [A] How can MotionCritic deal with sequences shorter than 60 frames and how does it perform ?

How is the input motion to MotionCritic normalized? Is the first frame translation at the origin? How is the global rotation set up? Are the first frame rotations set to the same rotation for each generation?

[B] How can MotionCritic be applied to sequences far longer than 60 frames, i.e. 1000 frames?

Table 1: NDMS and NPSS are flipped.

The evaluation methods against which motion critic is compared should be discussed in the related work, and not just in Table 1.

The authors should describe in how they apply the other metrics that they compare against, i.e. how they normalize the data, what kind of data representation/hyperparameters they are using.

**Questions:**

-

---

### Official Review · Reviewer_ud4c · 2024-11-11

**Soundness:** 4
**Presentation:** 3
**Contribution:** 3
**Rating:** 6
**Confidence:** 4

**Summary:**

This paper proposes a data-driven approach to align human motion generation with human perceptions. It introduces a large-scale dataset MotionPercept and a human motion critic model MotionCritic. The main contributions include providing a more accurate metric for motion quality assessment and showing that the critic model can enhance motion generation quality.

**Strengths:**

（1） The use of a large-scale human-annotated dataset and a critic model for automatic evaluation to align motion generation quality with human perceptions is a significant step forward.
（2） The experimental design is comprehensive. It evaluates the MotionCritic model on different data distributions and shows its generalization ability. Also, it tests the model as a training supervision signal and analyzes its impact on motion generation quality.

**Weaknesses:**

1. Table 2 shows, after 700 steps finetuning, the MotionCrotic decrease at 800 step. I am not sure about the trends, could you provide results with much much more steps to let us see the potential of designed Critic Supervision.

2. I think motion model trained on larger datasets HumanML3D and Motion-X is should be used to see Critic Supervision's potential.

3. Sec 4.3 could be with more detail and Algorithm 1 could be much simpler and clearer maintaining only necessary procedures.

**Questions:**

(1) Why not use the latest models, but MDM and FLAME released in 2022?

(2)Since currently, another two paradigms Noise De-masking and GPT-like Auto-regressive model show their potentials for motion generation task, how could this method of motion generation with critic model supervision be general to these methods?

(3) Why the step sampling range set as [700,900] in Sec 5.1? Is there any experiment support this setting, because I think from 700 - 900 step to get x0 directly results in motion with more artifacts, it might be help for those motion with obvious errors, but if you want to improve the motion quality use critic supervision as post-training, maybe it doesn't make much sense. If I am wrong, please explain your thoughts with some proofs.

BTW, I intially wanted to give 7 out of 10, but there is not this option. If authors can solve my concerns, I will raise the score.

---

### Meta-Review · Area_Chair_fnu1 · 2024-12-21

**Metareview:**

The authors present a new data-driven framework for evaluating and improving human motion generation. They propose (1) a large-scale human perceptual dataset, MotionPercept, that contains human preference annotations of synthetic motions; and (2) a human motion critic model, MotionCritic, which is trained on this dataset to capture high-level, subjective aspects of motion plausibility, smoothness, and overall realism from a human perspective. The paper’s second claim is that MotionCritic can serve not only as an evaluation tool but also as a supervision signal to improve the quality of generated motions. By fine-tuning existing motion diffusion models (e.g., MDM, FLAME) with critic scores, the authors show that generated motions become more physically correct and perceptually appealing.

Strengths of the Paper:

-- The authors construct a substantial dataset (MotionPercept) with more than 50k human preference labels, a valuable asset for the community and a strong foundation for learning perceptual scores.

-- MotionCritic provides sequence-level evaluation, enabling more nuanced assessments of motion plausibility compared to distribution-level metrics (e.g., FID).

-- The paper shows that using MotionCritic to fine-tune generative models yields improvements in motion realism, generalizing to different model architectures.

Weaknesses:

-- Since MotionCritic is trained on data from existing generators, it may overfit to their quality range and struggle if future models significantly surpass the baseline motions it was trained on.

-- Reviewers raise concerns about how the critic handles longer or more complex sequences (e.g., dancing, long-form actions). The dataset mostly uses shorter segments, potentially biasing the critic.

--The paper briefly mentions noise de-masking or auto-regressive approaches but does not provide extensive experiments showing how critic supervision might generalize to these newer techniques. Additional experiments on larger datasets and more training steps could strengthen claims.

Overall, the paper makes a valuable contribution by proposing an innovative perceptual metric and providing initial evidence of its benefits.  After carefully reading the paper, the reviews and rebuttal discussions, the AC agrees with the reviewers on recommending to accept the paper.

**Additional Comments On Reviewer Discussion:**

The weaknesses are described above. The authors have addressed most comments in rebuttal and the reviewers generally agree to accept the paper.

---

### Decision · Program_Chairs · 2025-01-22

Accept (Poster)